# Sirtuins-Mediated System-Level Regulation of Mammalian Tissues at the Interface between Metabolism and Cell Cycle: A Systematic Review

**DOI:** 10.3390/biology10030194

**Published:** 2021-03-04

**Authors:** Parcival Maissan, Eva J. Mooij, Matteo Barberis

**Affiliations:** 1Synthetic Systems Biology and Nuclear Organization, Swammerdam Institute for Life Sciences, University of Amsterdam, 1098 XH Amsterdam, The Netherlands; p.maissan@uva.nl; 2Systems Biology, School of Biosciences and Medicine, Faculty of Health and Medical Sciences, University of Surrey, Guildford GU2 7XH, Surrey, UK; e.mooij@surrey.ac.uk; 3Centre for Mathematical and Computational Biology, CMCB, University of Surrey, Guildford GU2 7XH, Surrey, UK

**Keywords:** Sirtuins, metabolic disorders, metabolic syndrome, cell cycle, pancreas, liver, brain, adipose tissue, muscle, heart

## Abstract

**Simple Summary:**

A vast number of molecules are involved in regulating metabolism in mammals. Among these molecules, Sirtuins play pivotal roles in the regulation of metabolism. Sirtuins are a family of seven members that are expressed in several tissues/organs and connect the inner and outer environment of the mammalian body to ensure a proper balance of metabolic activities. Deregulation of Sirtuins can be involved in a disturbed balance that is found in metabolic diseases such as obesity and cancer. The level and function of Sirtuins differ per tissue/organ and among mammals and shall be taken into account when envisioning administration of drugs that may affect Sirtuin activity. This systematic review provides an overview of the function of Sirtuins in six metabolic tissues/organs, and of the relevant processes that they regulate. Both healthy and metabolic disease conditions are discussed.

**Abstract:**

Sirtuins are a family of highly conserved NAD+-dependent proteins and this dependency links Sirtuins directly to metabolism. Sirtuins’ activity has been shown to extend the lifespan of several organisms and mainly through the post-translational modification of their many target proteins, with deacetylation being the most common modification. The seven mammalian Sirtuins, SIRT1 through SIRT7, have been implicated in regulating physiological responses to metabolism and stress by acting as nutrient sensors, linking environmental and nutrient signals to mammalian metabolic homeostasis. Furthermore, mammalian Sirtuins have been implicated in playing major roles in mammalian pathophysiological conditions such as inflammation, obesity and cancer. Mammalian Sirtuins are expressed heterogeneously among different organs and tissues, and the same holds true for their substrates. Thus, the function of mammalian Sirtuins together with their substrates is expected to vary among tissues. Any therapy depending on Sirtuins could therefore have different local as well as systemic effects. Here, an introduction to processes relevant for the actions of Sirtuins, such as metabolism and cell cycle, will be followed by reasoning on the system-level function of Sirtuins and their substrates in different mammalian tissues. Their involvement in the healthy metabolism and metabolic disorders will be reviewed and critically discussed.

## 1. Introduction

Since their discovery, proteins of the mammalian Sirtuin family have been in the spotlight because of their ability to regulate and influence pivotal cellular and molecular processes. These include aging, metabolism and disease development, among others. Sirtuins act through gene regulation, but also through post-translational modification of key proteins that are involved in metabolism, such as the AMP-activated protein kinase (AMPK), phosphoinositide 3-kinase (PI3K), the mammalian target of rapamycin (mTOR), the peroxisome proliferator-activated receptor (PPAR) proteins, and several cell cycle proteins such as cyclins and cyclin-dependent kinases (CDKs) and forkhead box (FOX) transcription factors. Through interaction and modulation of these proteins, Sirtuins influence and are affected by metabolism. However, Sirtuins, and their targets are not uniformly present in the metabolically active mammalian tissues. Sirtuin proteins (SIRT1-SIRT7) are differently localized, both within cells and between tissues, and some of their targets function differently in the various tissues. This variation is in large part due to changes in the expression of partners or targets of Sirtuins among cells and tissues. As a result of this differential expression, Sirtuins can behave heterogeneously between tissues, whilst drugs or compounds that activate Sirtuins are mostly administered at a systemic level. Consequently, despite a potential Sirtuin-mediated positive effect, a systemic treatment could lead to negative local effects.

The role of Sirtuins and most of their prominent substrates has been reviewed a few times. However, importantly, so far, no review has discussed the interaction between Sirtuins and their substrates in the regulation of metabolism and metabolic disorders in (several) mammalian tissues. Not all the molecular regulations involving Sirtuins’ function have been elucidated. However, such knowledge might provide insight into how metabolism is regulated in a healthy organism, as well as how metabolic disorders may occur due to Sirtuins’ deregulation. This information may possibly inspire development of new interventions. Therefore, this systematic review attempts to elucidate the specific role that interactions between Sirtuins and their prominent interaction partners play in different mammalian tissues, namely the heart, liver, adipose tissue, skeletal muscle, pancreas and the brain, in relation to metabolism and metabolic disorders.

## 2. Sirtuins and Metabolism

Sirtuins are members of the highly conserved family of SIR-TWO (SIRT), which can be found in virtually every organism, ranging from archaea to zebrafish [1]. Most prokaryotes encode at most two Sirtuins, whereas eukaryotes usually encode several Sirtuins that differ in their subcellular compartmentalization, from mitochondria, to nucleus or cytoplasm. Their shuttling between compartments is possible and depends on external factors. Eukaryotic Sirtuins also differ in their post-translational modification activities [2,3]. Besides the catalytic core, N- and C-terminal regions of Sirtuins vary in length, sequence and secondary structure, which are all not conserved. Sirtuins proteins are therefore divided in five different classes based on amino acid sequence homology, with the fifth class being exclusively prokaryotic [4].

Depending on organism and localization, Sirtuins can act as deacylases, desuccinylases, demalonylases, demyristoylases, depalmitoylases, mono-ADP-ribosyltransferases and, most importantly, deacetylases, or as a combination of these [5]. Most Sirtuins have the ability to deacetylate proteins and are members of the NAD+-dependent class III histone deacetylases (HDAC) protein family. Through deacetylating histones, these Sirtuins increase the amount of heterochromatin and modulate gene transcription through epigenetic regulation. Sirtuins are also able to target a plethora of non-histone substrates, mostly enzymes and transcription factors. The Sirtuins-mediated post-translational mo-difications have varying effects on protein activity depending on the specific substrate that is modified [6]. Protein deacetylation is the most common and important function of Sirtuins, with SIRT1, SIRT2, SIRT3 and SIRT5 having a higher deacetylation activity [2]. Acetylation influences protein stability by changing the electrostatic properties of proteins, thereby impacting on protein folding and the occurrence of specific protein-protein interactions. Furthermore, protein acetylation is important to control levels of a substrate, since deacetylated lysines can be ubiquitinated, thus marking a protein for degradation by the proteasome [7]. The cellular protein acetylation balance is closely regulated through interplay between histone acetyltransferase (HAT) and HDACs [8].

Sirtuins couple their enzymatic activity to the hydrolysis of NAD+. Sirtuin activity therefore results in two molecules, one of which is the unique metabolite 2-O-acetyl-ribose that has roles in cellular signaling [9]. The other molecule created through Sirtuin activity is nicotinamide, which can inhibit Sirtuin activity [2]. Sirtuins play roles in important biological processes such as aging, apoptosis, cell cycle, stress tolerance, and differentiation. Their dependency on NAD+, which plays a role in metabolic pathways, for their enzymatic activity, links them directly to the metabolism [10].

Regulation of Sirtuin activity occurs at several levels. Subcellular localization of Sirtuins is the first level of regulation; they can be active in either one or several cellular compartments, where they have to be located together with their specific substrate(s) [11]. Moreover, expression of Sirtuins is ubiquitous, but not at a uniform level across tissues [12]. Furthermore, availability of free NAD+ is of great importance to Sirtuin activity as without it no enzymatic activity would take place. Nicotinamide, created by Sirtuin activity, can act as a Sirtuin inhibitor (negative feedback) [13]. Finally, Sirtuin activity can also be regulated through post-translational modifications such as phosphorylation. For further information about Sirtuin regulation, we refer to another detailed review [14].

In humans, there are seven Sirtuin proteins. SIRT1 is the most studied among the Sirtuins and is known for its possibly lifespanextending effects and its role in healthy aging [15]. SIRT1 is mainly localized in the nucleus, but can also be found in the cytosol along with some of its substrates [11,16]. Transcriptional effects induced by SIRT1 result from its main localization in the nucleus. SIRT1 is also related to an increased mitochondrial metabolism, and protection against reactive oxygen species (ROS) [17]. SIRT1 has in fact been validated as a target against metabolic disease, and several SIRT1 activators have been developed, such as the well-known drug resveratrol. For further information about SIRT1 physiology, we refer to another detailed review [18]. SIRT2 is mainly found in the cytoplasm, but relocates to the nucleus during activation of the cell cycle [19], where it deacetylates tubulin during mitosis [20]. SIRT2 is most abundant in the brain, where it is involved in myelination [21]. SIRT2 exerts transcriptional effects through the deacetylation of transcription factors in the cytoplasm and directing them to the nucleus where they can, for example, influence the adipocyte differentiation, which indicates a role for SIRT2 in metabolism [22]. SIRT3 localization is a subject of debate [23,24], since it is mainly localized in mitochondria, but has also been found to localize in the nucleus under specific circumstances. Absence of SIRT3 results in mitochondrial hyperacetylation, thus indicating the role of SIRT3 as a major deacetylase. SIRT3 deficiency has been shown to accelerate the development of Metabolic Syndrome (MetS) [25]. SIRT4 is only found in mitochondria [11] and has ADP-ribosyltransferase activity as well as weak deacetylase activity on some substrates [26,27]. SIRT4 has roles in fatty acid oxidation and insulin secretion [26,28]. SIRT5 is mainly a lysine desuccinylase and demalonylase with possible weak deacetylation activity and can be found in both the mitochondria and the cytoplasm [29,30,31]. SIRT5 activity is known to influence diverse metabolic pathways, such as glycolysis and the tricarboxylic acid (TCA) cycle [30,32]. SIRT6 localizes both in the nucleus and in the endoplasmic reticulum, and has moderate acetylation capabilities as well as ADP-ribosyltranferase activity [33,34,35]. SIRT6 localizes to the nucleolus during the cell cycle, and is highly expressed during mitosis [36]. SIRT6 is known to be a regulator of glucose homeostasis [37]. SIRT7 is the least studied Sirtuin. It is found both in the nucleus and in the nucleolus, and is known as a deacetylase [38]. SIRT7 plays a crucial role as a regulator of RNA polymerase 1 transcription and is involved in both lipid metabolism and mitochondrial functioning, through glucose sensing and biogenesis of mitochondria, respectively. This evidence makes SIRT7 and its relation to metabolic diseases an interesting subject for further investigations, for its possible, similar roles of those of the most studied Sirtuins [39,40,41,42].

All mammalian Sirtuins thus play roles in regulating mammalian metabolism, albeit some more directly than others. Effects are exerted through the modulation of proteins acting as Sirtuin substrates. However, Sirtuins target many substrates, but not all of them are involved in regulating metabolism [6,7,43]. Besides influencing proteins directly through post-translational modifications, Sirtuins directly regulate chromatin through interaction with substrates such as transcription factors, epigenetic enzymes and histones. The net effects of this activity result in altered protein activity and levels available within the cell. Currently, prediction of the impact that metabolic changes have on protein acetylation, due to the interconnections between the metabolic network and pervasiveness of acetyl-CoA, is lacking. However, Sirtuins can secure their own NAD+ availability through the upregulation of AMPK, a protein that will be discussed below, resulting in Sirtuin activation during energy limitation.

In this review, interaction partners and substrates of Sirtuins are highlighted based on the available literature discussing Sirtuins and their role in metabolism in selected mammalian tissues.

## 3. Metabolism and Its (De)regulation

Metabolism is the ensemble of the life-sustaining chemical reactions within living cells of organisms. These reactions, catalyzed by enzymes, allow the organism to grow, reproduce, maintain their structure and respond to their environment. Currently, about 11,000 reactions and over 18,000 metabolites are annotated in the Kyoto Encyclopedia of Genes and Genomes (KEGG) [44]. However, not all organisms use the same set of metabolic reactions. For example, the human database contains about 6000 reactions, which is a considerable amount. However, the number of metabolites that humans can come into contact with—for example, through the food, environment, microbes or even consumables like cosmetics—is, according to the human metabolite database, as much as 114,010 [45]. Metabolism can also refer to other chemical reactions, such as the digestion and transport of metabolites both within and between cells, which is often referred to as intermediate metabolism. Metabolism is generally divided into two categories: (i) anabolism, the construction of cellular components such as nucleic acid and protein, and (ii) catabolism, the deconstruction of organic substances, including those toxic to the organism, through cellular respiration. In general, the construction of a metabolite consumes cellular energy, while deconstruction releases energy into the cell [46]. Chemical metabolic reactions are arranged into pathways in which one metabolite is transformed, through one or several steps, into the next metabolite by specific enzymes. These are crucial to the metabolism, as they lower the activation energy required for some reactions to be initiated, thus enabling organisms to drive reactions that would normally not occur or would be too slow to be effective. Enzymes thus act as catalysts, allowing reactions to proceed rapidly and regulating metabolic pathways in response to environmental changes or cellular signaling [46].

During glycolysis, glucose is metabolized into pyruvate with as net product four ATP molecules—proxy of energy transfer—and two NADH molecules—coenzyme carrying electrons from one reaction to another [47]. During fermentation, pyruvate and NADH are turned into oxidized NAD+, which is also required for the activity of all Sirtuins [48], and several other small molecules as end points of fermentation, some of which can be further fermented in other pathways [49]. Through anabolism, oxygen is created as a by-product and it is a powerful oxidant. Additionally, the reduced intermediates between oxygen and water are superoxide and hydrogen peroxide, and they also are potent oxidants. These reactive oxygen species (ROS) can induce significant damage to DNA, proteins and lipids [50]. For example, the rise of cyanobacteria and oxygen caused a massive extinction event called “the great oxidation” [51]. However, the presence of oxygen also created opportunities for life that survived. Oxygen enabled some of the by-products of fermentation, as well as pyruvate to be oxidized completely into CO_2_, allowing organisms to generate significantly more ATP than they could do before.

Nowadays, life on earth has evolved into a plethora of metabolic pathways. The pathways active in an organism organize which substances an organism will use as energy sources, and also which compounds are toxic when accumulated through either food or produced by other pathways. Some prokaryotes have evolved to use substances that are toxic to most other organisms as an energy source. An example thereof is a bacterium that can use hydrogen sulfide as an energy source [52]. As mentioned earlier, the substances used by organisms for nutrition are diverse; however, most organisms have a similar set of major metabolic pathways and their components [53]. The best example would be the carboxylic acids that are known as intermediates in the TCA cycle, which is present in some form in all known organisms using oxygen. These pathways arose early in the evolutionary history, evolved separately in several organisms, and have been conserved because of their high efficiency [54]. In prokaryotes, the TCA reactions are performed in the cytosol, whereas in eukaryotes those occur in mitochondria that not only act as the cells powerhouse, but also have roles in cellular signaling [55].

The breakdown of DNA and RNA occurs continuously in the cell. Pyrimidine breakdown products can be used in fatty acid synthesis and the TCA cycle. Ultimately, only ammonium will be left, which can enter the urea cycle. Purines are ultimately degraded into uric acid, a potent antioxidant. Pyrimidine and purine degradation intermediates can be salvaged to create more nucleotides when required [56]. Of note, many metabolic pathways exist, in which one central component can be transformed into one or several other components when needed.

The main metabolic pathways in mammals are: (i) glycolysis; (ii) TCA cycle, in which a series of reactions generate energy through oxidation of acetyl-CoA derived from carbohydrates, fats and proteins [56]; (iii) oxidative phosphorylation, in which mitochondria use their structure, enzymes, and energy released by oxidation of nutrients to generate ATP [56]; (iv) pentose phosphate pathway, parallel to glycolysis, that generates NADPH, pentose and ribose 5-phosphate, a precursor for synthesis of nucleotides; (v) amino acid synthesis pathways, in which amino acids are created from nitrogen and a carbon backbone that can be derived from either the glycolysis, the pentose phosphate pathway, or the TCA cycle [56]; (vi) urea cycle, in which the amino groups obtained from amino acid metabolism are removed and transformed into less toxic variants (in some organisms, the urea cycle can produce uric acid or ammonia) [56]; (vii) fatty acid β-oxidation, the catabolic process through which fatty acid molecules are broken down in mitochondria resulting in acetyl-CoA, which can enter the TCA cycle, and the redox cofactors NAD and FAD [56]; (viii) fatty acid synthesis, which results in the creation of fatty acids from acetyl-CoA, via malonyl-CoA [56]; (ix) gluconeogenesis, the metabolic pathway that results in the generation of glucose from non-carbohydrate carbon substrates like fat and proteins, which can be stored for quick energy access [56]; (x) glycogenesis, which many animals and some fungi use to store their glucose in the form of glycogen. Glycogen is the secondary energy storage for many animals, with the primary one being fat [56].

### 3.1. Metabolic Control

Metabolic systems are usually in a state in which the rate of reactions and the amount of metabolites vary only within a defined limit, i.e., a state referred to as homeostasis [57]. Such a tight regulation allows organisms to quickly respond to environmental signals [58]. Because any environment is rarely constant for a longer period of time, metabolic processes must be tightly regulated, as an imbalance may lead to improper functions, disease or even death.

Many environmental and intrinsic components play a role in the homeostatic regulation of an organism’s metabolism. Constant components can be either external, such as temperature or light, or internal, such as enzyme concentrations and enzyme properties. Variable components can be the metabolite concentration or the rate at which an internal reaction takes place [59]. The steady state of an organism is maintained by the variable components, which are the concentrations of metabolic substrates, as well as their interactions. Metabolites influence the rate at which enzymes consume them, and the resulting substrates are subsequently able to (indirectly) affect enzymatic rates through feedback loops [57]. Homeostatic control consists of a delicate balance of several components that regulate metabolic variables and of the changes in specific reactions parameters. For example, a regulatory molecule that changes the turnover rate of an enzyme will alter the steady state of the system [57]. The homeostatic control mechanisms that regulate the variables can be divided into (i) the receptor that receives information from the environment, (ii) the control center that processes the information, and (iii) the effector that initiates changes, when required. After sensing a change, a signal is transmitted to a monitoring component that sets the range at which the variable is maintained. This monitoring component will determine the response to the change sensed and, when needed, can transmit a signal. Any deviation will be corrected and, through negative feedback, the output of the system returns to its initial level [60]. Feedback is an important mechanism in controlling metabolism, as otherwise an organism would not know when to stop burning fuel and start storing energy, or vice versa.

There are multiple levels at which this feedback system acts. An intrinsic control exists, in which the pathway regulates itself in response to changes in substrate or product levels. This means that a decrease in product level can result in an increased enzymatic activity, or vice versa. This type of regulation is frequently allosteric in nature [61]. Some intrinsic pathways are reciprocally regulated, for example gluconeogenesis and glycolysis that lead to increased glucose levels; if both pathways were active, a significant waste of energy would occur. Therefore, the rate of glycolysis depends on the glucose concentration, whilst the rate of gluconeogenesis depends on the concentration of glucose precursors, such as lactate [56]. Some pathways can be activated simultaneously, resulting in a futile cycle. The main function of futile cycles is to generate heat, but it is also thought that these pathways play a role in metabolic regulation [62]. An extrinsic control also exists, in which cells of multicellular organisms change their metabolism upon receiving signals from neighboring cells, typically hormones or growth factors that are detected by cell surface receptors [63]. After receiving and transmitting these signals, the phosphorylation of proteins occurs, which can result in altered enzymatic function in a cell [64]. One of the most well-known extrinsic controls is the regulation of blood glucose levels by the hormones insulin and glucagon. Insulin release results in the uptake of glucose by cells, which is then converted and stored as either fatty acid or glycogen. Glucagon has the opposite effects of insulin, causing breakdown of glycogen into glucose, which is then transported out of the cell. Both insulin and glucagon are continuously secreted by the pancreas, to maintain a steady blood glucose level [65].

### 3.2. Metabolites Acting as Intracellular Signals

Most metabolites play a role in signaling, albeit often passively through sensing of metabolite levels by cells. An example thereof is sensing levels of glucose or ATP by brain cells or pancreatic β-cells [66,67]. In the latter, ATP senses glucose levels and in cooperation with ADP and Ca^2+^, induces insulin release [66]. However, several cellular metabolites, besides being a source for fuel or substrates for macromolecules, act as signaling molecules. One such metabolite is acetyl-CoA, which can actively participate in signaling and, in cooperation with the nucleo-cytosolic enzyme ATP-citrate ligase (ACL), epigenetic regulation [68]. Furthermore, metabolite-driven protein modification occurs. Proteins that serve as substrates or allosterically bind in signaling pathways can be covalently modified by metabolites to regulate their function. For example, some proteins were found to auto-acetylate when incubated together with the right acyl-CoA [69]. Most metabolites that have a function in signaling are different CoA’s performing functions such as acetylation; however, other metabolites such as the intermediary TCA cycle component fumarate can react with cysteine to form 2-succinyl-cysteine that negatively affects enzymatic activity [70], and to inhibit GAPDH in the presence of NADH.

Metabolites thus act as intracellular signals. Changes in the metabolism (and signa-ling) through dietary intervention can modulate health and lifespan. The most well-known and studied diets are the caloric restriction (CR) and ketogenic diet regimes. Both these regimes have been reported to act protective against development of metabolic disorders, neurodegeneration and cancer. However, many variations exist in the composition and duration of the diets, all with different reported effects. Diets will be not discussed here; for further information about the specifics of caloric restriction [71,72] and ketogenic diet [73], we refer to detailed reviews. Compounds exist that simulate these diets, such as resveratrol acting on SIRT1 [74]. Such similar compound(s) may be employed to fight against metabolic diseases, possibly without the need for relatively tough dietary interventions.

### 3.3. Metabolic Diseases

In major metabolic pathways, there are numerous proteins, hormones, metabolites and enzymes that influence metabolic homeostasis. The control switches for energy intake, storage or expenditure act in concert with the controls for food-restriction pathways active during starvation periods. These processes can in turn influence other cellular pathways such as growth, stress response, cell division and cellular differentiation. Because of the potential impact that dysregulation of the controls can have at a systemic level, these are fine-tuned and are evolutionary conserved [75]. However, when the metabolic balance is disrupted and cannot be recovered, severe metabolic and systemic consequences for the affected organism can occur in the form of a metabolic disorder.

Among the metabolic disorders, genetic inherited disorders involve congenital defects of the metabolism. These defects usually affect single genes responsible for the transcription of a particular enzyme; mutation results in a reduced or complete lack of substrate conversion. This missing functionality of this enzyme can subsequently lead to the toxic accumulation of upstream substances, lack of essential downstream substrates, or harmful alternative substrate metabolism. Considering the vast number of metabolic enzymes, many mutations can cause a possible disease. For patients of inherited metabolic diseases, treatment options are often available, such as dietary restriction, replacement of essential enzymes, or the modulation of certain pathways through introduction of novel compounds [76]. The other category of metabolic diseases is the non-communicable acquired metabolic diseases, which is prevalently influenced by risk factors such as gender, genetic background, environment, malnutrition, lifestyle and aging.

Several risk factors for developing a metabolic disease are already determined prior to birth. Gender can influence this development, through differences in body fat distributions and circulating sex hormones, as well as inherent differences in insulin sensitivity between sexes [77]. In addition, the genetic background and lifestyle of the parents of an individual can have a significant impact on the susceptibility for some risk factors. For example, Amerindian people, including European-Amerindian mixed groups such as Mexicans, seem to have a higher tendency to develop obesity at an early adult age and a higher incidence of diabetes mellitus (DM) and cholesterol gallstone formation as compared to people of European heritage [78]. Among European descent, it seems that a heritage of hailing from cold climates is correlated to a decreased prevalence of DM [79]. The biggest difference between these two European populations is their environment. The main link between the environment and the metabolic disorder is the thrifty phenotype hypothesis; this proposes that when a fetus is low on nutrients, it diverts energy from the development of metabolic organs (e.g., muscles, pancreas and liver) to instead use it for the brain. Besides developmental changes, the fetus will undergo permanent metabolic changes through epigenetic reprogramming, which will increase future survival chances in low-nutrient environments. If the fetus grows into a nutrient-rich environment, the chances of developing a metabolic disorder increase [80]. Proof in favor of the thrift-gene hypothesis is that the prevalence of DM is lower in sub-Saharan Africa, where poor fetal nutrition is often followed by poor postnatal conditions as compared to Western regions. Studies performed among children born during or shortly after the Dutch famine of 1944–1945 also indicated that these children had a significantly increased risk of developing a metabolic disease as compared to generations before them, although they all shared a similar genetic background [81]. The opposite of the thrifty phenotype, i.e., the feast phenotype, has also been proposed. For this phenotype it is hypothesized that obese parents can affect the epigenetic programming of their offspring, just as starved parents will [82]. The cause of differences in susceptibility between ethnicities is not known; however, theories exist that propose that some cultures started investing more time and effort into fewer offspring and transitioning from a lifestyle that favors the use of muscles to a lifestyle that favors the use of the brain. Since the transition to less offspring is usually coupled to an increased use of the brain, it is hypothesized that this behavioral switch is coupled to insulin resistance [83]. It could, however, also be envisioned that some ethnicities have incorporated some of the metabolic changes caused by fetal nutrition shortage into their genome as the result of permanent hunter-gatherer culture and are therefore more susceptible to metabolic diseases. A possible explanation for why colder-climate heritages are more resistant to developing DM is that they developed higher blood glucose levels, as glucose is a natural cryoprotectant [84], and may therefore have evolved an adapted insulin response. This line of thought is supported by the higher prevalence of type 1 diabetes observed in colder climates, which also results in higher blood glucose levels [85]. Lifestyle and diet also have a major effect on development of a metabolic disease, since eating in excess and low exercise are risk factors, as are drinking alcohol and smoking tobacco [86]. Finally, the aging factor is characterized by deterioration of homeostatic maintenance processes over time, leading to functional decline and increased risk for diseases as a result of resistance to signaling, together with physiological and hormonal changes. Thus, age-related metabolic changes must be taken into account when prescribing medicines to elderly patients, to ensure that the homeostatic balance is not disrupted [87].

Cancer is considered a metabolic disease, since cancer cells usually have a unique metabolism, known as the Warburg effect, and an increased ROS resistance [88] to cope with their microenvironment. Moreover, many cancers have common risk factors that are also found to be correlated with metabolic diseases [89]. Besides DM and cancer, there are many different types of metabolic diseases, such as heart diseases, liver diseases and vascular disease, and risk factors can have an accumulative effect on the acquisition of one or more of these diseases. Since many metabolic pathways are intercalated, metabolic diseases rarely manifest by themselves. The association between different metabolic diseases, such as diabetes, high blood pressure (hypertension) and obesity, is referred to as metabolic syndrome (MetS) [90].

### 3.4. Metabolic Syndrome

MetS refers to both the concept and diagnosis of a syndrome defined by the observed association of a wide range of pathologies such as clinical obesity, dyslipidaemia, hypertension, DM, fatty liver disease, glucose intolerance, microalbuminuria, chronic inflammation, and cancer. MetS and its related diseases are important risk factors for cardiovascular diseases and morbimortality, and are the underlying cause of many deaths and billions in healthcare costs every year, whilst the prevalence of MetS is steadily increasing [91]. As mentioned earlier, many risk factors associated with the development of metabolic diseases exist that play a role in the susceptibility to develop MetS. A sedentary lifestyle and a diet rich in fats and carbohydrates are thought to be the biggest underlying causes, as they lead to a quick accumulation of body fat, which—when at levels of clinical obesity—is the cornerstone of MetS [92,93]. Fat tissue consists of many cell types, but mostly preadipocytes, which can differentiate into adipocytes when required, and adipocytes themselves. Adipocytes are active endocrine cells that are able to rapidly respond to nutrient intake through both hypertrophy and hyperplasia [94]. When obesity causes the progressive enlargement of adipocytes, blood supply to these cells will be reduced, since cardiac output and the proportion of blood flow directed to adipose tissue are not increased. In addition, the sheer size of adipose cells limits oxygen availability, especially when distant from capillaries, thus resulting in local hypoxia [95]. Hypoxia results in tissue necrosis and infiltration of macrophages, which both lead to an increased secretion of a group of cytokines by adipose tissue called adipokines [96].

Adipokines integrate endocrine, paracrine and autocrine signaling, to modulate several metabolic processes such those involved in immune response, ROS stress, insulin signaling and energy metabolism. Some adipokines, such as the tumor necrosis factor α (TNFα), are proinflammatory proteins that induce inflammation, which can spread to surrounding tissues, resulting in chronic inflammation associated with MetS [97]. TNFα also results in adipocyte apoptosis, and is an inhibitor of the insulin receptor substrate 1 (IRS1) pathway [98]. By inhibiting insulin signaling and causing adipocyte cell death, TNFα promotes the release of free fatty acids (FFAs) from adipocytes into the bloodstream and reduces adipocyte glucose and FFA uptake. TNFα also reduces insulin signaling in muscle cells [99,100]. Elevated FFA levels also result in inhibition of insulin-stimulated glucose uptake and inhibition of glycogen synthesis in muscle cells. Compensatory insulin secretion will eventually result in insulin resistance [100]. However, FFAs are responsible for around 30–50% of basal insulin secretion, and not all obese individuals develop DM, since they can still compensate for the FFAs-mediated insulin resistance with FFAs-mediated insulin secretion. A genetic compound is therefore suspected to be behind the reason that some obese individuals develop DM; because they are unable to compensate for the altered signaling or because they suffer more quickly from receptor fatigue [100]. In the liver, insulin signaling is also suppressed by FFAs, resulting in the suppression of glycogenesis and in an increased risk for fatty liver disease [101]. Chronic elevated levels of FFAs also impair pancreatic β-cell function, resulting in desensitization and suppression of insulin secretion [102]. Adiponectin, an adipokine, is negatively correlated to fat tissue mass, and has positive effects on whole-body metabolism, such that it can reverse some of MetS symptoms. Adiponectin stimulates glucose utilization and β-oxidation through the activation of AMPK, and can enhance hepatic insulin function as well [103,104]. Adiponectin is in essence acting as an insulin sensitizer, and has been found to reduce the effects of hyperglycemia and insulin resistance when administered to people with diabetes [103,105]. Thus, in obese individuals, more glucose and FFA are circulating in the bloodstream. To cope with this scenario, more insulin needs to be secreted, but in the long term, this increase would cause insulin-receptor desensitization and a subsequent insulin resistance. Insulin resistance is thought to be the second cornerstone of MetS [106].

An insulin-resistant phenotype is characterized by elevated fasting glucose levels, hyperglycemia, impaired glucose metabolism and normal insulin levels, no longer resulting in an adequate response by targeted tissue. This will make pancreatic β-cells produce extra insulin in an attempt to normalize glucose levels, thus resulting in hyperinsulinemia. Insulin resistance and hyperinsulinemia are therefore rarely isolated symptoms [107]. In the short term, hyperinsulinemia can compensate for insulin resistance and allow maintenance of normoglycemia, but insulin sensitivity is heterogeneous between tissues, so some insulin-sensitive tissues may exhibit elevated insulin-mediated activity [108]. The resistance to insulin in one tissue but increased activity in another is also one of the clinical manifestations of MetS. Pancreatic β-cells will compensate for the increased insulin resistance by expanding total β-cell mass, as well as enhancing β-cell function [109,110]. β-cell compensation to insulin resistance is a proliferative response of the cells to growth factor-mediated signaling, and it requires nuclear exclusion of the forkhead box protein O1 (FOXO1) [111]. Whilst the compensatory mechanisms of β-cells might solve the problem to a certain extent, it fails in the long term. At some point, the β-cells will suffer from glucotoxicity, circulating fatty acids, leptin and cytokines, with the stress resulting in β-cells to go into apoptosis. Apoptosis in the presence of cytokines can, sometimes, also result in auto-immunity, a scenario in which the body will further attack β-cells [112]. When β-cells decrease, the individual has effectively contracted diabetes mellitus.

Since hyperglycemia increases oxidative DNA damage, it increases the risk of developing several types of cancer [113]. Moreover, insulin resistance is often accompanied by other clinical symptoms of MetS, such as cardiovascular disorders. After insulin binds the insulin receptor, which is a tyrosine kinase, activation of the receptor results in downstream tyrosine phosphorylation of the substrates in two parallel pathways. These are the MAP-kinase pathway and the PI3K/AKT pathway. Insulin resistance results in a desensitized PI3K-Akt activation and in an excess of MAP-kinase activity. The respective increase of MAP-kinase activity impairs glucose metabolism and increases MAP-kinase-regulated insulin resistance gene expression [114], as well as continued endothelin production and mitogenic stimulation of vascular smooth muscle cells [115,116]. At the same time, decreased PI3K/AKT activation results in a decreased glucose transporter type-4 (GLUT4) translocation, thus impairing glucose uptake in fat and muscle tissues [117] and decreasing endothelial NO production, leading to decreased endothelial functioning [118]. Thus, the imbalance between MAP-kinase and PI3K/AKT pathways caused by insulin resistance can result in vascular abnormalities that are predisposing for cardiovascular diseases, such as atherosclerosis.

Also, insulin resistance causes dyslipidaemia, which is characterized by “the lipid triad” consisting of elevated triglyceride and LDL levels and decreased HDL levels [119]. Insulin resistance can result in the lipid triad in several ways. Adipocyte lipolysis is suppressed by insulin, so insulin resistance results in an increased lipolysis and FFAs levels, with FFAs being substrates for triglycerides in the liver [120]. Triglycerides in turn regulate the production of VLDL and apolipoprotein B (ApoB), which is the primary apolipoprotein of VLDL and LDL [121]. ApoB is broken down by insulin through the PI3K pathway, which—as previously mentioned—is impaired during insulin resistance [122]. Insulin also stimulates the production and secretion of lipoprotein ligase that plays a major role in VLDL clearance, so during insulin resistance, lipoprotein ligase activity is impaired [123,124]. The triglycerides in VLDL can be exchanged for cholesteryl esters with HDL through the cholesterol-ester transport protein, with triglyceride HDL being easier for hepatic lipases to clear, resulting in less HDL in circulation for the reverse cholesterol transport chain [125]. In the liver of insulin-resistant individuals, FFAs flux and triglyceride synthesis are increased, with both FFAs and triglyceride in turn being secreted as VLDL. Dyslipidaemia is thus the result of increased circulating triglycerides, increased VLDL production, and reduced VLDL clearance. VLDL will be further reduced to LDL and remnant lipoproteins, which can both promote atheroma [126,127]. Dyslipidaemia is often associated with oxidative stress and endothelial dysfunction, further reinforcing the inflammatory and cardiovascular aspects of MetS. Insulin resistance and hyperglycemia can also increase the activity of the renin–angiotensin–aldosterone (RAA) system through angiotensin II, resulting in an increased blood pressure [128]. In addition, sympathetic nervous system activation can further contribute, by increasing sodium reabsorption and vasoconstriction [129]. Moreover, adipocytes are able to produce aldosterone in response to angiotensin II signaling, thus further contributing to the development of hypertension [130]. Many metabolic pathways are interconnected and so are metabolic diseases. Insulin plays a central role in energy homeostasis and metabolism, and influences several cellular processes such as the cell cycle. Exposing cells to insulin induces mitosis and DNA synthesis, highlighting the relevance of insulin in cell cycle regulation [131]. Several pathways involving insulin have been found to regulate cellular proliferation and cell cycle progression, indicating a direct link between metabolism and the cell cycle.

### 3.5. Linking Metabolism to the Cell Cycle

In order for cells to proliferate, growth and cellular division shall respond to signaling. Whether or not cells enter the cell cycle can be viewed as a fundamental metabolic issue, because the available nutrients and the metabolic status must be reviewed to ensure sufficient biomass to be produced, to support production of a daughter cell. Cells may either enter the cell cycle or remain in a quiescent state, awaiting favorable conditions. Once conditions are appropriate, a cell must activate the appropriate metabolic response to initiate cell division and steer it to completion. Genome-wide expression studies have suggested that expression of metabolic enzyme is synchronized with the cell cycle phases [132]. It has been shown in budding yeast that mitochondrial enzymes needed for glycolysis and oxidative phosphorylation were induced in early G1 phase [132], and a cell cycle-dependent regulation of genes required for nutrient uptake and amino acid synthesis [133]. Further research on this organism discovered oscillations in oxygen consumption corresponding to a yeast metabolic cycle (YMC). These metabolic cycles appear to correspond to periodic expression of half of all yeast transcripts, and each phase of the metabolic cycle appears to be linked to a phase of the cell cycle [134]. Besides oxygen oscillations, cycles of intracellular concentrations of several metabolic compounds such as amino acids, nucleotides, NADP, NADPH and acetyl-CoA, were also observed [135]. Oscillation of these compounds reflects both rebuilding of cellular components after division, and metabolic changes during the cell cycle [135]. Altogether, this evidence indicates a direct link between the cell cycle and the metabolism, with building of a new cell requiring a myriad of metabolic reactions.

## 4. The Cell Cycle and Its (De)regulation

To ensure that cell division is initiated only when a cell has grown sufficiently, has enough nutrients available and has successfully replicated its DNA, a number of checkpoints exist through which a tight regulation of cell cycle phases is guaranteed. Cell division is mainly regulated through timely gene expression and oscillatory activity of members of the catalytic cyclin-dependent kinase (CDK) family of serine/threonine kinases, which activity is driven by the regulatory cyclin proteins that form with CDKs binary Cyclin/CDK complexes. CDKs were first described in budding yeast [136]. In this organism, cdc28—later renamed Cdk1—is the only CDK regulating progression throughout the cell cycle, and its null mutant can be complemented by human CDKs, indicating the evolutionary conservation between CDKs across species [137]. Around 20 CDKs have been identified in the mammalian cells; however, most of them are not directly active in driving cell division and only 11 have been studied extensively [138]. Five among these 11 CDKs, i.e., CDK1, CDK2, CDK3, CDK4, and CDK6, have direct roles in regulating cell cycle timing. Moreover, only CDK1 is required for cell viability, albeit a bit slower than a wild type cell [139]. CDKs require a certain phosphorylation pattern to function properly: CDK7 is a positive regulator of CDK activity [140], while specific phosphatases are negative regulators [141].

CDK requires its binding partners, the cyclins, to exploit the kinase activity on substrates. Cyclins do not have any enzymatic activity, but when a cyclin protein forms a complex with a CDK, the active site of the latter is exposed. Cyclins also confer subunit specificity and localization as extra regulatory roles [142,143]. Conversely, the inhibition of Cyclin/CDK complexes occurs via cyclin-dependent kinase inhibitors (CKIs), with p27Kip1, p21Cip1 and INK4 being the most important regulators [144]. Not all CDKs are committed to one specific cyclin as a partner, but can instead form complexes with several cyclins in different concentrations at the same time during cell cycle progression. Different cyclin partners provide different possibilities regarding binding and localization [138]. Despite the ability of most CDKs to bind different cyclins, the former are usually found in complex with cyclins of a certain family, which can be considered to be the CDK’s preferential cyclin partner [138]. Cyclins that bind to the human CDKs are A1, A2, B1, B2, B3, C, D1, D2, D3, E1 and E2. Different cyclins have different localization patterns: Cyclin B localizes to the cytoplasm, while Cyclin A, Cyclin C and Cyclin D localize to the nucleus. The loss or mutation of a single cyclin can usually be compensated by another cyclin, mostly from the same family [138].

The number of CDK proteins within cells during the cell cycle remains fairly constant [145], however their activity is controlled through a timely, oscillatory expression and the breakdown of different cyclins as well as of CKIs during the various cell cycle phases [146,147]. Not all cyclins are equally expressed, and their timely synthesis drive cell cycle events [148]. For the cell cycle to progress from one cell cycle to another, cyclins have to be degraded by the cyclosome that targets them to the proteasome. Active CDKs can phosphorylate a large number of substrates, which has been shown to be over 300 for both budding yeast and mammalian cells [43,149]. Through interaction with many substrates, CDKs are able to coordinate the cell cycle phases in a timely fashion.

### 4.1. Cell Cycle Control

The eukaryotic cell cycle is controlled by a highly conserved network which general features are found from yeast to human [150]. The beginning of a cell cycle occurs when a cell is in interphase after a recent division. In interphase, a cell grows and receives nutrients and, when conditions are favorable, mitogenic signals will be transmitted through pathways such as Wnt, Shh and EGF. These signals activate the transcription of Cyclin D and of the transcription factor Myc [151,152,153]. Myc upregulates the transcription of Cyclin D, through positive feedback, while downregulating transcription of the p21Cip1 inhibitor of CDK2 complexes [154,155]. p21Cip1 has a dual role as both a CDK2 inhibitor and an assembly factor for the Cyclin D/CDK4,6 complexes, therefore acting also as a positive regulator of the cell cycle [156]. To commit to cell division, phosphorylated Cyclin D/CDK4,6 complexes monophosphorylate the retinoblastoma protein (pRB). pRB phosphorylation occurs in concert with pRB deacetylation by SIRT1, since acetylation protects pRB from phosphorylation events [157]. pRB prevents cell cycle progression by binding proteins of the E2F transcription factor family. After mono-phosphorylation, pRB changes the binding to E2F sufficiently for allowing progression from G0 to G1 phase, but not enough to fully release E2F [158]. The pRB-E2F complex is known to recruit HDAC to the chromatin of E2F sites, allowing for heterochromatin formation through which transcription of cell cycle promoting factors is halted, further slowing down the cell cycle [159].

Myc also promotes transcription of Cyclin E and inhibits p27Kip1, which binds to and inhibits Cyclin E/CDK2 complexes [160]. Myc inactivates p27Kip1 by repressing its transcription factor FOXO3A [161]. The Cyclin E/CDK2 complex, when activated, contributes to increased pRB phosphorylation; a hyperphosphorylated pRB releases E2F, which initiates the transcription of its target genes, such as Cyclin E, through positive feedback [162,163]. As a result, pre-replication complexes are assembled that require Cyclin E activity for DNA replication [164]. Cyclin E/CDK2 complexes can also phosphorylate E2F proteins, to increase their interaction with co-factors, further increasing the positive feedback loop [165]. Furthermore, Cyclin E/CDK2 can phosphorylate, thereby promote, the degradation of the CDK inhibitors p21Cip1 and p27Kip1 [166,167]. Upon DNA damage, Cyclin E/CDK2 has been shown to initiate apoptosis through phosphorylation of the transcription factor FOXO1 [168].

At the G1/S transition, E2F induces transcription of Cyclin A, which replaces Cyclin E to form Cyclin A/CDK2 complexes [169]. At the same time, Cyclin E is degraded through p53-induced activation of the ubiquitin-mediated proteasome. Cyclin A/CDK2 activity results in accumulation of Cyclin B through phosphorylation of the transcription factor FOXM1 together with phosphorylation and subsequent rearrangement of the anaphase-promoting complex (APC) [170,171]. Cyclin A/CDK2 also plays a role in the formation and separation of the centrosome through phosphorylation of CP110 [172]. Cyclin A inhibits the assembly of new pre-replication complexes through the phosphorylation of proteins involved in DNA replication, such as CDC6 [169]; when the concentration of Cyclin A/CDK2 complexes reaches a threshold, DNA replication will initiate [164]. This ensures that G1 phase ends before S phase begins, preventing DNA from being replicated more than once per cell cycle. At the end of S phase, Cyclin A associates with CDK1. The Cyclin A/CDK1 complex regulates the firing of origins of replication [173], and triggers degradation of proteins through phosphorylation. Finally, Cyclin A/CDK2 activates and coordinates Cyclin B/CDK1 complexes to the nucleus, where they work together to initiate nuclear envelope breakdown and chromatin condensation [174,175,176]. At the start of the prometaphase, Cyclin A is removed after acetylation through ubiquitin-mediated degradation [177].

The Cyclin B/CDK1 complexes are now responsible for progression through mitosis and continue to be transcribed whilst mitosis occurs [178]. To exit mitosis, Cyclin B/CDK1 complexes are removed by ubiquitination-mediated APC degradation. Cyclin B/CDK1 complexes stimulate APC activation at the end of mitosis [179]. At this point, the cell cycle is complete, and the newborn cell will remain in interphase while waiting for Cyclin D to accumulate, to begin a new cell cycle.

### 4.2. Linking the Cell Cycle to Metabolism

Cell cycle and metabolic processes are conserved in many organisms, and connections between these have been highlighted for mammalian cells [54,180]. At the G1/S transition, the E2F transcription factor family regulates several factors involved in metabolism [181]. Specifically, E2F1 regulates energy homeostasis and mitochondrial function in muscle and brown fat tissue; the CDK4-Rb-E2F axis represses mitochondrial oxidative metabolism, and E2F1 deficiency results in an altered energy metabolism [182]. Conversely, an increase of mitochondrial ROS levels causes increased E2F1 levels [183]. Other studies link the CDK-Rb-E2F axis to metabolic parameters such as skeletal muscle metabolism, pancreatic β-cell size, white adipose tissue, and insulin signaling. This evidence indicates that cell cycle regulators can directly influence glucose homeostasis [184]. The relevance of upstream CDKs in metabolism is indicated by a study were mice deficient for the two CDK inhibitors, p21Cip1 and p27Kip1, developed obesity significantly faster as compared to wild type mice [185]. These mechanisms are likely to be conserved in humans, as genome wide association studies (GWAS) have indicated through identification of certain loci of CDK inhibitors that are correlated to DM susceptibility [186].

Cell division, hypertrophy and apoptosis are related to the many metabolic diseases associated with MetS, and are controlled by the regulation of the cell cycle [187]. As previously mentioned, several cell cycle regulators play roles in metabolic processes such as glucose homeostasis and function of the adipose tissue. Considering that obesity and insulin resistance are causes of MetS, a misfunctioning of the cell cycle may contribute to the development of metabolic disorders. Studies have found that high-fat diets (HFDs) and obesity result in altered cell cycles in pancreatic β-cells and muscle tissues [188,189]. Furthermore, diabetic patients have abnormal angiogenesis [190], and kidney cell cycles are dysregulated in diabetic nephropathy [191]. Thus, besides a link observed between the cell cycle and metabolism, a link may be envisioned between relevant cell cycle regulators and the pathogenesis of metabolic disorders.

## 5. Sirtuins Substrates Involved in Regulating Metabolism

### 5.1. AMPK—The Metabolic Swich

The AMP-activated protein kinase (AMPK) is activated by metabolic stresses that inhibit ATP synthesis or accelerate ATP consumption, and is an integrator of energy signa-ling in regulating whole-body energy balance [192]. AMPK is seen as a metabolic master switch at both the cellular and systemic level, with the main functions being nutrient sensing, and control of metabolism and of mitochondrial biogenesis [193]. AMPK is expressed in many tissues related to metabolism, such as the brain, muscle, liver, adipose tissue and heart, where it regulates different signaling pathways related to metabolism. Once activated, AMPK switches off processes that consume ATP whilst activating catabolic pathways [194,195].

AMPK activation has numerous downstream effects, such as the stimulation of fatty acid oxidation and ketogenesis in the liver, inhibition of lipolysis and lipogenesis in adipocytes, stimulation of glucose uptake in muscles, and modulation of insulin secretion by pancreatic β-cells [196]. AMPK is also involved in the inhibition of protein synthesis through the mTOR pathway, and the inhibition of cell cycle progression through the p53/p21Cip1 axis [197]. In addition, AMPK interacts with other cell cycle proteins, such as FOXO and SIRT1, that are also involved in longevity and metabolism [198,199]. Speci-fically, SIRT1 is required for AMPK activation [199], and this interaction in turn enables the improvement of mitochondrial functioning upon resveratrol administration. In addition, AMPK phosphorylates FOXO, thereby enhancing transcriptional activity [198]. Subsequently, changes in energy metabolism are induced and stress resistance is increased. Finally, AMPK interacts with SIRT1 to improve mitochondrial functioning. The role of AMPK as nutrient sensor seems to be coupled to the completion of mitosis, through sti-mulation of growth factors and proto-oncogenes, including PI3K, AKT and ERK [195,200]. Furthermore, interactions between CDK and AMPK subunits have been shown to control some AMPK functions [201]. For further information about AMPK structure, regulation and general activity, we refer to another detailed review [202]. AMPK has been implicated in several metabolic diseases, and modulation of its activity is of interest as potential the-rapeutic intervention. In this context, it is relevant to consider the proteins that influence, and are influenced by AMPK, such as the Sirtuins.

### 5.2. The PI3K/AKT/mTOR Pathway

Phosphoinositide or phosphatidylinositol 3-kinases (PI3Ks) are a family of intracellular signal transducer enzymes that are capable of phosphorylating a phosphatidylinositol after being activated [203]. PI3Ks are divided into three main classes, based on structure, regulation and lipid substrate specificity, and a fourth class of PI3K-like proteins was defined later [204]. Several forms of phosphorylated phosphatidylinositol are observed, depending on the class of PI3K that mediates phosphorylation [205]. PI3K products can bind to and activate a wide number of cellular target proteins that function as second messengers. Subsequently, a complex signaling cascade is initiated that alters several cellular activities, such as proliferation, development, differentiation and glucose homeostasis [206]. Downstream substrates of PI3K can activate AKT. The role of PI3K/AKT related to insulin response and glucose metabolism has been described earlier; importantly, activated AKT regulates a number of important players in the cell cycle: it inhibits p27Kip1 and localizes FOXO to the cytoplasm [207,208]. PI3K can be antagonized by phosphatase and tensin homolog (PTEN), which dephosphorylates the phosphatidylinositol [209].

Another major target of AKT is mTOR: AKT activates mTOR through the phosphorylation and inactivation of AMPK. In fact, AMPK phosphorylates and deactivates tuberous sclerosis complex 2 (TSC2), the negative regulator of mTOR, so that AKT is responsible for the activation of mTOR [210]. mTOR proteins serve as core components of two protein complexes, mTOR complex 1 and mTOR complex 2, which are able to regulate different cellular processes, among which cell growth, proliferation, survival, protein synthesis and transcription. mTOR complexes act through integration of input from upstream factors such as insulin or other growth factors [211]. mTOR also plays roles in sensing cellular energy, nutrient and oxygen levels [212]. The mTOR pathway plays critical roles in mammalian metabolism and physiology through activity in many tissues, including liver, muscle, brown adipose tissue and the brain, and has been found to be dysregulated in diabetes and obesity. For further information about the role of mTOR in metabolism and proliferation, we refer to other recent detailed reviews [213,214].

### 5.3. PPAR Transcription Factors

Peroxisome proliferator activated receptors (PPARs) are a group of transcription factors and nuclear receptors. Endogenous ligands of PPARs are fatty acids and eicosanoids [215,216]. When PPARs bind its ligand, transcription of target genes can be either increased or decreased depending on the particular target and signal. PPARs play roles in cellular differentiation and metabolism of higher eukaryotes [217]. All PPARs are expressed in the adipose tissue, with the PPARγ2 isoform being expressed in at high levels in the adipose tissue, and different PPARs can be found in metabolic tissues such as the liver, muscle, kidney, brain, pancreas and intestines [217,218]. PPARs regulate the expression of several genes involved in metabolic processes, which are also linked to development of obesity, hyperlipidemia, diabetes and, consequently, of MetS, through regulating the metabolism of lipids, carbohydrates and amino acids.

PPARs are activated by several ligands, with one of the most relevant ones being the unsaturated free fatty acids (FFAs) [219]. PPARs heterodimerize with the retinoid X receptor (RXR) before to bind target genes. PPARs can be activated through acetylation and deactivated through deacetylation by the transcriptional coactivator p300 and SIRT1, respectively. Conversely, activated PPARγ is able to inhibit SIRT1 activity [220]. In essence, because of their activation by FFAs, PPARs act as nutrient sensors. Three types of PPARs exist: alpha (α), delta (δ) and gamma (γ). PPARγ1 has a broader expression pattern involving different tissues [217]. Depending on the tissue, activation of PPARs can have different effects due to the presence of tissue specific co-factors, expression of downstream target genes, and post-translational modifications.

PPARα’s main function is to regulate lipid metabolism in the liver [221,222]. PPARα is mostly expressed in muscle tissue, in the liver and, to a lesser extent, in the kidneys, heart, intestines and adipocytes [223]. In a fasted state, PPARα expression is activated by glucocorticoids [224]. PPARα transcription has also been reported to be increased by the satiety hormone leptin [225], whilst insulin-dependent phosphorylation also plays a role in PPARα activation [226]. PPARα promotes intake and usage of fatty acids through genes involved in fatty acid binding and transport, and by both mitochondrial and peroxisomal fatty acid β-oxidation. PPARα is also required for ketogenesis, the main adaptive response to fasting. Lack of PPARα has been linked to impaired fasting responses, low levels of ketone bodies, fatty liver, hypoglycaemia, and hypoinsulinemia [221,227]. PPARα regulates processes, mainly through altering expression of hundreds of target genes [222]. In summary, PPARα regulates energy homeostasis through fatty acid catabolism and gluconeogenesis. PPARα agonists, such as fenobrate, are successfully employed in clinical settings against hyperlipidemia, non-alcoholic fatty liver and dyslipidemia [228,229].

PPARγ is mainly expressed in adipocytes, and regulates adipocyte differentiation and lipid storage; however, it also has roles in lipid biosynthesis and insulin sensitivity [230,231]. Mice lacking the PPARγ gene are unable to become obese, even on HFDs [232]. PPARγ plays a role in insulin sensitivity through regulating the secretion of several factors, such as adiponectin and leptin [233]. PPARγ can be regulated by insulin [230] and, during clonal expansion, PPARγ transcription is activated through E2F1 [234]. PPARγ is also regulated through several post-translational modifications. Specifically, interactions of PPARγ with CDK5 have been associated to an increased insulin sensitivity [235], and those with CDK9 and CDK4 have been associated to an increased adipogenesis [236,237]. Interestingly, Cyclin G2, an unconventional cyclin that is usually upregulated during apoptosis or growth suppression, has been found to bind PPARγ and stimulate adipoge-nesis [238]. PPARγ also interacts with peroxisome proliferator-activated receptor gamma coactivator 1-alpha (PGC1α), the master regulator of mitochondrial biogenesis, which mediates the interaction of PPARγ to multiple transcription factors. Both AMPK and SIRT1 have been found to interact with PGC1α [239]. Currently, PPARγ antagonists, such as rosiglitazone, are successfully employed clinically against insulin resistance; however, these drugs have side effects because of their low specificity [240].

Compared to other PPARs, PPARδ is considerably less studied. It is ubiquitously expressed, but found most abundantly in muscle cells [241]. PPARδ is a potent inhibitor of the transcriptional activity of PPARα and PPARγ. PPARδ inhibition works partly through its association with co-repressors, as well as through binding to the peroxisome prolife-rator response element [242]. PPARδ has also been implicated in the regulation of the burning of fatty acids in muscle and fat tissue through controlling expression of fatty acid uptake and β-oxidation genes [243]. PPARδ activation induces a metabolic shift in the liver, thus reducing glucose output and increasing insulin sensitivity [244]. The role of PPARδ in tumor growth is currently investigated, however its role remains unclear. Therefore, any therapy involving PPARδ is still controversial [245].

In summary, PPARγ contributes to energy storage by enhancing adipogenesis and fat storage, whilst PPARα and PPARδ mainly facilitate energy expenditure. PPARs have been found to interact with Sirtuins, AMPK, mTOR and FOXO, pointing to the relevance of the former to link the cell cycle with metabolism [246,247,248,249].

### 5.4. Forkhead Box Proteins

The forkhead box (FOX) protein family is named after their DNA binding domain, which is a winged helix of roughly 100 amino acids called the forkhead domain. This domain is highly conserved within the FOX family and across species [250]. Other parts of FOX proteins are highly diverse and can for example encode regions involved in activation or repression of FOX functionality [251]. In humans, up to 50 FOX proteins have been discovered, divided into 19 subclasses based on phylogenetic analysis of the forkhead domain [252]. A number of FOX proteins have roles in development, and their mutation can result in several developmental abnormalities that can even lead to death before, or shortly after birth [253,254]. Others play roles in several critical biological processes such as cell growth, proliferation, differentiation and longevity, and their deletion may result in disease [253,254].

Among the mammalian FOX proteins, the FOXO subfamily (FOXO1, FOXO3, FOXO4, and FOXO6) has received attention because of the impact on longevity and metabolism, conserved across different species [255]. FOXO1 and FOXO3 are widely expressed among different tissues, whereas: FOXO4 is mostly expressed in muscle, kidney, and colon tissue; and FOXO6 is primarily present in liver and brain [256]. Of note, FOXO1 regulates adipocyte differentiation [257] and has a role in hepatic glucose production [258]. FOXO transcription factors have been implicated in glucose production, reduction of insulin section, food intake, and skeletal muscle degradation. During insulin resistance occurring in obesity, FOXOs are actively causing diabetes and hyperlipidemia. For further information about the role of FOXO in in metabolic regulation, we refer to another detailed review [259]. FOX proteins are regulated through a complex combination of post-translational modifications that can alter their localization and function, among which the deacetylation activity mediated by Sirtuins. For further information about the functioning of the FOXO code, we refer to another detailed review [260]. Not much information is available about the interplay between FOX proteins and Sirtuins. FOX homolog proteins in budding yeast, the transcription factors Fkh1 and Fkh2 have been found to associate to Sir2 (SIRT1 ortholog) in the transcriptional regulation of mitotic cyclins, repressing cell cycle progression [261]. In mammals, an interaction between FOXO and SIRT1 has been described in vascular growth, maintenance and aging [262].

Since most Sirtuins and their partners are involved in major metabolic processes, below we provide a detailed information of their function in the different metabolic tissues: pancreas, liver, brain, adipose tissue, muscle and hearth (see Appendix A for details about systematic search strategy, study eligibility and study selection procedure). This overview highlights current and future therapies that are and may be pursued to treat metabolic diseases.

## 6. The Role of Sirtuins in Metabolic Tissues

### 6.1. Pancreas

SIRT1 is present in the cytoplasm of pancreatic β-cells but absent in exocrine pancreatic cells, and it regulates insulin secretion [263]. Specifically, SIRT1 deacetylates the insulin receptor-2 (IR2), affecting its susceptibility for phosphorylation by insulin [264]. Pancreatic SIRT1 and PGC1α expression increases during a caloric restriction (CR) diet and decreases during high-fat and high-fructose diets. Similarly, FOXO3a level decreases in high-caloric diets, but no change is detected during CR, whereas PPARγ level increases during a high-fat diet but decreases during either CR or a high-fructose diet [265]. Hyperlipidemia (i.e., a condition where too many lipids or fats, such as cholesterol and triglycerides, accumulate in the blood) has a negative effect on SIRT1 expression, increases localization of FOXO1 to the nucleus, and reduces insulin secretion. Resveratrol is able to reverse these effects through activation of SIRT1 [266]. Considering that: (i) FOXO1 has a negative effect on β-cell function, (ii) β-cells overexpressing FOXO1 seem to have decreased amounts of glucagon and insulin secretion, and (iii) inhibition of FOXO1 seems to protect pancreatic cells against stress-related apoptosis, it is likely that deacetylation of FOXO1 by SIRT1 and the reversal of nuclear localization are responsible for the observed rescue by resveratrol [266,267,268]. An increased secretion of glucose-stimulated insulin through resveratrol, and the subsequent activation of SIRT1, have also been confirmed in human pancreatic islet cells [269].

FOXO activates the cell’s pro-apoptotic program. This seems to occur, in part, through the regulation of the nitric oxide (NO) response, which either induces cell death or cell repair. Activated SIRT1 modulates FOXO1, preventing apoptosis and activating cellular repair [270]. FOXO1 plays a positive role in the proliferation of β-cell mass. The glucoincretin hormone glucagon-like peptide 1 (GLP-1) inhibits SIRT1 activity by decreasing the NAD+/NADH ratio and directly reducing SIRT1 expression. Consequently, FOXO1 acetylation increases, resulting in an increased β-cell proliferation in the pancreas of mice [271]. FOXO1 also has a role in protecting pancreatic cells against hyperglycemia-induced oxidative stress [272]. In diabetic mice, activation of SIRT1 also increases β-cell mass, indicating that both a reduction and an increase of SIRT1 activity can result in β-cell proliferation depending on the circumstances, since control mice had no increases in β-cell mass when SIRT1 was overexpressed [273]. In diabetic mice, CR upregulates SIRT1 expression and increases insulin sensitivity while decreasing apoptosis rates in pancreatic cells [274]. Moreover, SIRT1 partially protects β-cells in rats from lipotoxicity [275].

In mice, overexpression of SIRT1 in pancreatic β-cells results in an improved glucose tolerance and enhanced insulin secretion in response to glucose. In fact, microarray data revealed that SIRT1 expression was able to regulate genes involved in expression of insulin [276]. However, as the mice were aged between 18–24 months, glucose-stimulated insulin excretion through SIRT1 is blunted. This study suggested that this observation is due to a decreased NAD+ synthesis in aged individuals, and that a NAD+ enhancement may be a potential therapeutic intervention to prevent age-associated metabolic disorders [277]. Pancreas-specific SIRT1 deficiency results in lower insulin secretion by β-cells, but is not accompanied by hyperglycemia, suggesting that a compensatory mechanism may exists [278]. An earlier study, however, indicated that SIRT1 regulates pancreatic islet formation and differentiation through deacetylating FOXA2, and that pancreatic disruption of SIRT1 resulted in a progressive hyperglycemia together with glucose intolerance and a lack of insulin [279]. The difference between these two studies is that in the former, the deficiency is at 3% of control levels, whereas in the latter, a complete SIRT1 deletion was achieved. These data indicate that even extremely low levels of SIRT1 suffice to prevent hyperglycemia. Since loss of pancreatic SIRT1 results in a reduced expression of mitochondrial genes for metabolic coupling, it may be envisioned that a low level of SIRT1 is able to promote gene expression necessary for a glucose response, albeit a weaker one [280].

Further evidence that SIRT1 plays a positive role in the pancreas was found in rats, where the activation of SIRT1 through a drug promotes β-cell regeneration through sti-mulation of AMPK, which mimics the effect of CR [281]. The anti-malaria drug Artesunate was shown to protect pancreatic cells from cytotoxic damage by upregulation of SIRT1 and reduction of NO production, which—as mentioned earlier—together with FOXO1 has an apoptotic effect. This drug may be effective against the inflammatory damage of cells in MetS [282]. Administration of GABA was also shown to reduce pancreatic apoptosis in a SIRT1-dependent manner, by increasing NAD+ levels [283]. Low doses of resveratrol were shown to support β-cell proliferation in a high-glucose environment in porcine pancreatic stem cells, and to increase SIRT1 expression while decreasing levels of the tumor protein p53. However, higher doses of resveratrol had opposite transcriptional effects and even promoted the apoptosis of β-cells [284]. Furthermore, Fucoidan, derived from brown algae, has been shown to ameliorate impaired insulin production by reducing pancreatic β-cell death in a SIRT1-dependent manner [285]. Altogether, this evidence shows that, when manipulating Sirtuins through drugs, patients should be carefully monitored.

No information for the metabolic role of SIRT2 in pancreatic function, besides in cancer, can be retrieved. However, according to the human protein atlas, low levels of SIRT2 RNA have been detected in the pancreas, thus warranting future research on a potential role for SIRT2 in pancreatic function [286].

SIRT3 has an anti-inflammatory role. Its decreased expression in pancreatic β-cells of diabetic patients likely contributes to β-cell impairment due to inflammation [287]. SIRT3 deficiency increases the vulnerability of pancreatic β-cells to palmitate-induced oxidative stress in mice [288]. Overexpression of SIRT3 attenuates effects of β-cell dysfunction in cells grown in low concentrations of the fatty acid palmitate, increases insulin secretion and decreases β-cell apoptosis. Thus, SIRT3 seems to protect against lipotoxicity, thus against β-cell dysfunction in obese individuals [289,290]. However, another study showed that cells grown on high glucose/palmitate, thus undergoing glucolipotoxicity, had fewer death rates when either SIRT3 or SIRT4 were knocked down. This evidence suggests that nicotinamide has a protective effect against glucolipotoxicity due to its inhibitory action on Sirtuins [291]. This discrepancy may suggest that protective effects of SIRT3 could change depending on the amount of macronutrients. Further research is required, to clarify this aspect.

SIRT4 is highly expressed in pancreatic β-cells and negatively regulates insulin secretion, with SIRT4-knockout mice having higher circulating levels of insulin. Levels of active SIRT4 decrease during CR, resulting in an increased insulin secretion [26,48,292].

The role of SIRT5 in the pancreas is ambiguous and is described in only two studies. One study revealed that SIRT5 is downregulated in the pancreas of mice suffering from type 2 diabetes mellitus (T2DM), and plays a role in maintaining β-cell mass and function in glucolipotoxic conditions, retaining insulin secretion [293]. However, the other study showed that SIRT5 is upregulated in humans with T2DM and suppresses β-cell prolife-ration *in vitro* through the inhibition of PDX1 transcription via H4K16 deacetylation. Since PDX1 is involved in β-cell activity and formation, SIRT5 seems to play a role in T2DM through its role as a histone deacetylase. The latter study also showed that downregulation of SIRT5 promotes insulin secretion [294]. The discrepancy between these two studies may result from the difference between human and mice and warrants further research into SIRT5 as a therapeutic target against T2DM.

In the pancreas, SIRT6 deacetylates FOXO1. SIRT6 knockout in β-cells of mice impairs glucose-stimulated insulin signaling. As a result, mice become glucose intolerant, while development of islets is not influenced [295]. SIRT6 deficiency leads to an increased acetylation of H3K9Ac and H3K56Ac at the promotor of *Txnip*, resulting in a time-dependent increase of the corresponding protein and in a subsequent reduced glucose tolerance and insulin secretion. Stimulation of SIRT6 could therefore be beneficial to prevent T2DM [295]. Islet cells without SIRT6 also show decreased ATP levels and oxygen consumption. The impairment can be rescued by ectopic expression of SIRT6, indicating a role in the pancreas for SIRT6, together with FOXO1, in insulin signaling and glucose homeostasis [296,297].

Currently, no studies regarding the role of SIRT7 on pancreatic function can be found. However, according to the human protein atlas, low levels of SIRT7 RNA, medium levels of SIRT7 protein in the exocrine glands, and low SIRT7 levels in the islets of Langerhans were detected, thus warranting future research on a potential role for SIRT7 in pancreatic functions [286].

An overview of the role of Sirtuins in the pancreas is presented in Table 1. Altogether, research is relatively scarce as compared to other tissues. SIRT2 and SIRT7 have not been investigated yet, and research on SIRT4 and SIRT5 is scarce, with SIRT4 being promising against hyperinsulinemia. SIRT1 generally has a positive effect, and SIRT3 may have a positive effect but further research is required to understand the underlying mechanisms. SIRT6 interacts with FOXO, and may be involved in insulin signaling. No studies about Sirtuins interacting with mTOR or AMPK exist, although a number of studies describe the role of these proteins, individually, in the pancreas. Only one study on SIRT1 and PPARγ can be retrieved. Moreover, one study about SIRT1 and p53 deacetylation can be retrieved, related to cancer [298], which may suggest a role for SIRT1/p53 in metabolic disorders. An interaction between forkhead transcription factors and Sirtuins in the pancreas is discussed in a few papers, while the role of FOXO has been described in many studies. The pancreas plays an important role in cellular metabolism; thus, these interactions may be relevant as targets for therapies against metabolic disorders.

### 6.2. Liver

In the liver, SIRT1 can act as a nutrient sensor via NAD+, favoring gluconeogenesis and increasing glucose output during fasting, while suppressing glycolysis, through deacetylation of PGC1α. Increased glucose concentrations decrease SIRT1 protein levels, whereas pyruvate increases them. SIRT1 protein levels are likely regulated in a post-translational manner, since SIRT1 mRNA levels do not vary during the process [299,300]. The role of SIRT1 in glucose homeostasis is possibly also mediated through PPARγ, which is increasingly acetylated when SIRT1 activity decreases during alcohol consumption [301]. PPARγ expression in the liver has been linked to liver regeneration, but also to the development of fatty liver disease, among others, and is downregulated during CR [302,303,304]. Hepatic SIRT1 deficiency in mice was shown to increase hyperglycemia, oxidative stress and insulin resistance through impaired AKT/mTOR signaling [305]. FOXO1 interacts with PGC1α to regulate glucose homeostasis [306], and SIRT1 has deacetylates FOXO1, which promotes the transcription of gluconeogenic genes [307].

SIRT1 also plays a role in liver lipid metabolism through: (i) activation of AMPK, which decreases fatty acid synthesis and stimulates fatty acid oxidation [308], and (ii) positive regulation of PPARα, which is responsible for the adaptive response to fasting and stimulates fat uptake and fatty acid β-oxidation in the liver [309]. PPARα is also implicated in increasing plasma HDL and decreasing circulating triglycerides, which are both associated with a decreased incidence of cardiovascular diseases [310]. A cell cycle inhibitor, called p16Ink4, was found to play a role in hepatic glucose homeostasis and hepatic lipid metabolism by suppressing fatty acid oxidation. Vice versa, ablation of p16Ink4 increased fatty acid ablation through the AMPKα2/SIRT1/PPARα pathway [311,312]. In glucose homeostasis, p16Ink4 suppressed the PKA/CREB/PCG1α pathway [312]. Thus, cell cycle mechanisms may bridge SIRT1 and PCG1α pathways, with p16Ink4 being potentially involved in T2DM development by modulating these metabolic pathways. Adiponectin plays a role in the activation of AMPK/SIRT1 and PPARα. Specifically, a decrease in adiponectin during obesity, and subsequent activity loss of affected pathways, contribute to MetS development. Conversely, an increase in adiponectin mediates bad liver histology [313,314]. AMPK also negatively regulates the PI3K/AKT/mTOR pathway that is responsible for an increased lipogenesis in the liver, even during insulin resistance, acting through FOXOs [315,316].

Reduction of SIRT1 deacetylase activity increases HFD-induced non-alcoholic fatty liver disease (NALFD) development [317]. SIRT1 is activated during fasting by PLIN5, which promotes autophagy and decreases inflammation levels, thereby maintaining hepatocyte homeostasis [318]. Alternate day fasting also increased SIRT1 levels in the liver of diabetic mice and reduced insulin resistance, inflammation, obesity and prolonging of insulin signaling [319]. Furthermore, a substantial decrease in SIRT1 levels has been observed in aged mice compared to young mice, with older mice exhibiting an altered lipid metabolism [320]. Considering that most individuals develop MetS at an older age, age-dependent alternations of lipid metabolism and protein expression could exacerbate disease progression. SIRT1 levels are downregulated in humans with NAFLD and in HFD-fed mice, resulting in a decreased β-oxidation and in disease progression. SIRT1 expression is known to ameliorate NAFLD [321]. Activating SIRT1 also has an effect on glucose tolerance and lipid accumulation in the liver of diabetic mice [322]. When SIRT1 is deleted in liver cells, PPARα signaling is impaired and β-oxidation decreases, whereas the upregulation of SIRT1 achieves the opposite effect [309]. A CR diet can upregulate SIRT1 and partially ameliorate NAFLD [323]. Furthermore, upregulation of SIRT1 attenuates NAFLD and inflammation caused by a HFD in mice [324]. Although CR increases SIRT1 in NAFLD and HFD decreases SIRT1 levels in the diseased condition, in healthy livers, the opposite effect occurs [323,325,326,327]. However, because this research was conducted in rats, further research is needed to clarify why disease conditions can cause a different response to a specific diet.

Metformin was shown to alleviate fatty liver through increasing SIRT1-mediated autophagy [328]. SIRT1 also plays a role in cholesterol efflux, through deacetylation of the liver X receptor (LXR) in the nucleus, thereby activating it [329]. During diet-induced obesity, p53 can be dysregulated, resulting in its inhibition by SIRT1 and accumulation of lipids in the liver. This scenario promotes oxidative stress, which leads to increased p53 levels that can activate downstream genes with roles in insulin resistance [330]. Inhibition of p53 attenuated these effects in HFD-fed mice [331]. Exercise results in an increased SIRT1 expression that leads to an attenuated inflammation and metabolic disfunction in the liver of diabetic mice [332]. A few compounds have shown an effect on lipid accumulation. Resveratrol increases SIRT1 expression in the liver, thereby ameliorating lipid droplet accumulation during the process of HFD-induced steatosis in mice [333]. Furthermore, resveratrol (i) facilitates SIRT1 activity; (ii) possibly regulates genes responsible for lipid oxidation; and (iii) prevents development of steatohepatitis by protecting against endoplasmic reticulum stress [334]. Folic acid attenuates HFD-induced hepatic steatosis in rats [335]. Furthermore, an increased expression of genes related to de novo lipogenesis, β-oxidation and lipid uptake occurred, likely through SIRT1-dependent activation of PPARα [335]. Maslinic acid, found in olive peels, protects against NAFLD through the AMPK/SIRT1 pathway, reducing liver weight and lipid accumulation in mice [336]. Vitamin K1 regulates the AMPK/SIRT1 pathway, lowering insulin resistance and fasting glucose levels in diabetic mice [337]. Several other compounds that attenuate NAFLD and lipid metabolism in the liver through the AMPK/SIRT1/PGC1α pathway have been described [338,339,340,341]. For example, LB100 was shown to ameliorate NAFLD in HFD-induced obese mice, by reducing hepatic FFA accumulation and injury. LB100 acts through the AMPK/SIRT1 pathway by upregulating both SIRT1 and AMPK levels [342]. Furthermore, three compounds were shown to reduce FFA levels by increasing SIRT1 levels and acting through the AMPK/SIRT1 pathway: Y-mangostin in liver cells *in vitro*, Apple Polyphenol Extract in HFD-induced NAFLD mice, and liraglutide in HFD-induced diabetic mice. Finally, liraglutide has also been shown to increase insulin sensitivity [343,344,345]. In addition, Phoenixin 14—a neuropeptide—induced weight loss and liver mass reduction and had an anti-inflammatory role partly due to activation of the AMPK/SIRT1 pathway [346]. This compound could be used against NAFLD. Interestingly, probiotic treatment increased SIRT1, PGC1α and PPARα levels in liver tissue of HFD-fed mice and upregulated genes involving bile synthesis, cholesterol homeostasis and fatty acid oxidation [347]. This evidence indicates the potential connection between the gut-brain axis and metabolic diseases.

Maternal HFD in mice reduces SIRT1 expression and increases the incidence of liver disorders in the offspring. On the other hand, overexpression of SIRT1 in the offspring reduces body weight, increases glucose tolerance and hepatic insulin sensitivity, thus indicating that SIRT1 has a role in fetal metabolic reprogramming [348]. When fed an HFD, prenatal stress in pregnant mice resulted in lower SIRT1 levels and increased glucose and insulin levels compared to wild type [349]. In addition, females gained more weight compared to males. Interestingly, overexpression of SIRT1 in the HFD-parent attenuated metabolic disorders in the mice offspring, reversing glucose intolerance and normalizing fat morphology with increasing SIRT1 levels [350]. Offspring also had lower birth weights and reduced adiposity and hyperlipidemia.

Inhibition of SIRT2 was shown to suppress hepatic fibrosis [351]. SIRT2 overexpression attenuated hepatic steatosis in high-fat/high-sucrose (HF/HS) fed mice [352]. SIRT2 was downregulated in insulin-resistant mice livers, accompanied by an increased ROS generation and mitochondrial dysfunction. SIRT2 overexpression restored insulin sensitivity, by counteracting oxidative stress and mitochondrial dysfunction [353]. SIRT2 can promote hepatic glucose uptake and tolerance in HFD-fed mice [354]. Specifically, decreased SIRT2 levels reduce glucose uptake, and SIRT2 knockdown results in an impaired glucose uptake and tolerance in the liver.

SIRT3 is a prominent regulator in CR adaptation, through the deacetylation of proteins involved in diverse pathways of metabolism and mitochondrial maintenance [355]. SIRT3 is involved in ketogenic pathways, which are activated upon fasting [356]. During CR, SIRT3 promotes fatty acid oxidation and the urea cycle [357]. Fatty liver is associated with a decreased SIRT3 expression, which results in hyperacetylation of mitochondrial proteins that normally protect against lipotoxicity upon nutrient excess [358]. In bovines, NALFD has also been associated with reduced SIRT3 levels, while upregulation of SIRT3 mitigates triglyceride deposition by modulating genes involved in hepatic lipid metabolism [359]. SIRT3 knockout in HFD-fed mice results in an increased steatosis accompanied by a reduced expression of PPARα and genes involved in fatty acid oxidation [360]. In addition, SIRT3 deficiency resulted in triglyceride accumulation in the liver. However, an HFD reduces only SIRT3 levels in males [361]. Contrarily, modulation of the SIRT3/FOXO1 pathway through salvianolic acid B was shown to improve NAFLD in rats, by reducing oxidative stress [362]. SIRT3 protects hepatocytes of mice from oxidative injury by enhancing ROS scavenging and mitochondrial integrity [363]. Increasing SIRT3 levels by inhibiting adenyl cyclases results in deacetylation of mitochondrial proteins, thereby protecting them against potential ischemic damage [364]. The anti-ROS effects of SIRT3 are likely due to its deacetylase activity on PPARγ and FOXO1, to promote mitochondrial biogenesis and increase oxidative defense [362,363]. Decreased SIRT3 expression during NAFLD could be due to palmitic acid—levels increase during obesity—that downregulates SIRT3, thus resulting in oxidative stress and apoptosis in the liver [365]. Decreased SIRT3 levels are also associated with an impaired intestinal permeability [366]. The drug berberine was found to ameliorate NAFLD in rats through the AMPK/SIRT3 pathway, indicating that this axis plays an important role in NAFLD [367]. Another compound, the protocatechuic acid, was found to increase SIRT3 expression and protect against NALFD in rats. SIRT3 knockout completely reverses this effect; also SIRT3 deacetylates the protein ACSF3, thus modulating fatty acid metabolism [368]. Polyphenols extracted from blueberries were found to activate the AMPK/SIRT3/PGC1α pathway in human and rat liver cells [369]. This results in an improved hepatic mitochondrial dysfunction and redox homeostasis, which mitigate liver steatosis and alleviate inflammation, thus preventing further damage. Thus, phytotherapy treatment has the potential to be used in NALFD prevention and treatment. In mice, treatment with Shizophyllan resulted in an increased SIRT3 expression and activation, which in turn resulted in less complications from a lipodystrophy model in mice [370]. This compound increases mitochondrial metabolism and mediates mitochondrial resuscitation through SIRT3.

SIRT3 can also activate hepatic fat catabolism, and its levels increase during fasting [371]. Furthermore, SIRT3 promotes lipophagy, protects liver cells against lipotoxicity, and can suppress lipogenesis through activation of AMPK [372]. The increased SIRT3 expression during fasting is due to increased levels of PGC1α and PPARα during fasting [373]. PGC1α, via SIRT3 and SIRT5, plays a role in the regulation of the urea cycle in the liver through the regulation of carbamoyl phosphate synthetase 1 (CPS1) [29,374]. CPS1 is critical for clearance of the excess of ammonia, which is produced upon metabolization of proteins during fasting, or of a high protein diet [375]. During fasting in response to glucagon, the activity of these proteins increases to help detoxify the increased amounts of ammonia [29,374].

SIRT4 inhibits fatty acid oxidation and mitochondrial gene expression in the liver, and inhibits expression of SIRT1 as well as repressing PPARα [28,376]. When SIRT4 is knocked down, an increase of SIRT1 expression and fatty acid oxidation is observed [28]. In pigs SIRT4 plays a role in glutamate synthesis through the mTORC1/SIRT4 pathway when downregulated by leucine, whereas SIRT4 knockouts did not respond, indicating a role for SIRT4 in the metabolism of different nitrogen sources [377]. A weight loss drug called 3-Iodothyronamine (T1AM) is responsible for reducing SIRT4 levels in the liver of mice [378].

SIRT5 is activated in liver mitochondria upon fasting, or during a prolonged high-protein diet. In the liver, SIRT5 has also been implicated in the regulation of ketogenesis and β-oxidation, as these processes decrease when SIRT5 levels are lacking [30]. SIRT5 is downregulated in NAFLD [379], whereas its overexpression attenuates hepatic steatosis and other metabolic abnormalities in obese mice [380]. SIRT5 is upregulated through PGC1α and PPARα during fasting, but downregulated by AMPK. Of note, overexpression of AMPK in mouse hepatocytes increases expression of SIRT1, SIRT2, SIRT3, and SIRT6 [373].

In the liver, SIRT6 is regulated by SIRT1 and FOXO3a, and can negatively regulate glycolysis, triglyceride synthesis and fat metabolism. Deletion of SIRT6 can reverse these effects and contribute to NAFLD [381]. Upregulation of SIRT6 in mice—through SIRT1-mediated deacetylation—results in a negative regulation of glycolysis, triglyceride synthesis and fat metabolism, due to SIRT6-dependent deacetylation of histone H3K9 [382]. In agreement with this evidence, SIRT6 is downregulated in human livers affected by NAFLD [382], suggesting that SIRT6 plays a role in NAFLD through its histone deacetylase activity, and in cholestatic livers [383]. In addition, SIRT6 knockout mice predispose to developing NAFLD, thus indicating a protective role for SIRT6 in the liver [384,385]. SIRT6 has been also shown to be downregulated in obese rats, but it is upregulated after energy restriction-induced weight loss, together with an increase of oxidative stress markers [386]. The drug rolipram, which reduces ageing-related adipose disposition, was shown to increase SIRT6 levels in livers of mice [387]. Moreover, the drug T1AM increases SIRT6 levels, whereas it decreases SIRT4 levels, as well as reversing obesity through metabolic reprogramming [378]. In response to this recent mouse model study, *in vivo* experiments are planned to further develop and refine the anti-obesity applications initially observed with T1AM. FOXO3a and SIRT6 also play a role in lowering LDL levels [388]. Moreover, SIRT6 overexpression protects against aberrant glucose homeostasis, and increases insulin sensitivity [389]. Finally, SIRT6 activates PPARα to promote fatty acid β-oxidation in the liver, thereby reducing liver fat content [390]. In obese rats, exercise improves SIRT6-mediated insulin signaling and ameliorates insulin resistance [391]. In mice with a cholestatic liver, activation of SIRT6 reduces liver damage and fibrosis, through repression of the nuclear estrogen-related receptor γ (ERRγ) [383]. In humans with cholestasis, lower SIRT6 levels have been reported, thus the upregulation of SIRT6 might be a potential therapeutic intervention.

SIRT7 positively regulates the nuclear receptor TR4/TAK1, which is involved in lipid metabolism, by controlling the ubiquitin pathway that leads to its degradation. Activated TR4 increases fatty acid uptake and triglyceride synthesis. SIRT7 knockout mice were resistant against developing HFD-induced NAFLD [392]. In this study, SIRT7 has been shown to protect against NAFLD, with HFD mice lacking SIRT7, developing a NAFLD-like of humans. In the same study, it has been shown that lack of SIRT7 increases trigly-ceride content and fatty liver without obesity, as the SIRT7-deficient mice were leaner as compared to controls. On the other hand, overexpression of SIRT7 protected the mice from NAFLD by reducing ER stress through deacetylation of the histone H3K18A [393]. SIRT7 is targeted to promotors of ribosomal genes through interaction with the transcription factor Myc. It may be possible that, through its histone deacetylase activity, SIRT7 alleviates ER stress by counteracting Myc activity after deacetylation of the histone [393]. Further research into the mechanisms of SIRT7 function in the liver is needed.

An overview of the role of Sirtuins in the liver is presented in Table 2. SIRT1 and SIRT3 exert a positive effect, while SIRT4 a negative effect. SIRT2 seems to have a negative effect, however further research is needed to uncover the underlying mechanism of action. SIRT5 seems to play a role during fasting, but the exact mechanisms are currently unknown. However, SIRT5 downregulation during NAFLD may indicate a positive role, given that Sirtuins with a positive influence are generally downregulated during NAFLD.

SIRT6 seems to be tied to SIRT1 function, but has an overall positive effect. The role of SIRT7 remains ambiguous. A role for AMPK, PPARs, forkheads, p53 and the mTOR pathway in relation to Sirtuins has been investigated in the liver. PGC1α is extensively discussed in literature, but its connection to PPARγ is not investigated. However, since it is found to play a role in alcoholic fatty liver, it may also play a role in NAFLD, although these diseases mostly differ epidemiologically [394].

### 6.3. Brain

In the brain, the hypothalamus plays a critical role in sensing peripheral signals through a region called the arcuate nucleus, which has a modified blood–brain barrier that allows nutrients, hormones and other signaling molecules to enter [395]. Within this nucleus, two distinct populations of neurons exist, namely the anorexigenic POMC neurons and the orexigenic AgRP neurons. These neurons are characterized by the excretion of a specific neuropeptide that has a potent effect on energy homeostasis [396]. FOXO1 and SIRT1 are expressed in both AgRP and POMC neurons, with FOXO1 being localized in the nucleus in the fasted and in the cytoplasm in fed condition [397]. FOXO1 regulates food intake through the inhibition of POMC neurons. In the latter, leptin and insulin signaling converge on PI3K/AKT, which phosphorylates and subsequently removes FOXO1 from the nucleus.

When the AKT pathway is blocked, FOXO1 signaling and subsequent food intake increase, resulting in weight gain [398]. Fasting increases levels of SIRT1 in the hypotha-lamus, resulting in an increase of FOXO1 activity through deacetylation and decrease in POMC expression, which stimulates hyperphagia in mice [399]. Inhibition of hypothalamic SIRT1 activity reverses the fasting-induced decrease of FOXO1 acetylation, resulting in an increased POMC and a decreased AgRP expression, thereby in decreased food intake and weight gain [399]. In agreement with these observations, SIRT1 inhibition in mice with diet-induced obesity results in increased energy expenditure and a decreased body weight [400]. It was found that SIRT1 overexpression increases energy expenditure in POMC neurons through an increased adipose tissue activity, and decreases food intake in AgRP neurons together with an increased leptin sensitivity [401]. These phenotypes were observed in aging rats, but not in rats on an HFD [401]. Neuronal SIRT1 also promotes the response to dietary restriction through activation of the dorsomedial and lateral nuclei of the hypothalamus by augmenting ghrelin. Consequently, an increase of body temperature and physical activity in mice is observed, which can stimulate foraging behavior [402]. SIRT1 is required for the homeostatic defense against diet-induced obesity in mice [403]. Overexpression of SIRT1 is able to rescue obesity-induced insulin resistance in POMC neurons of mice where insulin-resistant nuclear FOXO1 has become constitutive [404]. Finally, neuronal SIRT1 negatively regulates insulin signaling in the hypothalamus, lea-ding to insulin resistance that spreads to peripheral tissues, besides affecting the brain [405]. Moreover, SIRT1 deficiency increases insulin-stimulated phosphorylation of both AKT and FOXO1, and SIRT1 inhibition in neurons results in an increased central and systemic insulin sensitivity [405].

The reported effects of SIRT1 in the hypothalamus are in general opposite to those that SIRT1 confers to peripheral tissues. In light of this, SIRT1 activators that do not penetrate the blood–brain barrier might be preferable. The SIRT1 activator resveratrol may have a detrimental effect on brain mitochondria at serum levels several times lower than would be necessary for peripheral mitochondria [406]. Conversely, in obese mice with a reduced expression of SIRT1 in the hippocampus coupled with memory deficits, activating SIRT1 through resveratrol preserves the hippocampus-dependent memory [407,408]. Moreover, 4-hydroxyestrone, an estrogen metabolite with little estrogenic activity, was shown to increase SIRT1 activity in the brain, resulting in deacetylation of p53 and subsequent protection against oxidative damage in the rat hippocampus [409]. A decreased SIRT1 activity in the hippocampus during obesity is caused by circulating palmitic acid, which lowers SIRT1 activity through NAD+ depletion [410]. Another SIRT1 agonist, named SRT1720, reduces cognitive decline in T2DM but results in an increase in body weight [411]. Mice overexpressing SIRT1 in the forebrain exhibit an increase in fat accumulation, impaired glucose tolerance and motor defects [412].

The SIRT1/p53 axis plays a role in food intake through the stomach-derived peptide ghrelin, which activates hypothalamic AMPK. Inhibition of SIRT1 blocks the orexigenic action and leaves ghrelin-related release of growth hormones intact [413]. SIRT1 is necessary for starvation-induced autophagy, as mice lacking SIRT1 are unable to initiate autophagy when starved [414]. SIRT1 is also required for the physical activity increases observed during CR in mice [415]. Of note, neuronal SIRT1 has the potential to slow down Alzheimer’s disease during CR [416], and fasting enhances the SIRT1 mediated deacetylation of PPARγ, which activates PGC1α and regulates the rate-limiting enzyme for amyloid beta (Aβ) generation [417]. In rats, CR attenuates ischemic brain injury; furthermore, knockdown of SIRT1 removes the neuroprotective effect, indicating that SIRT1 is involved in inducing ischemic tolerance [418]. It probably does this through deacetylation and inhibition of the hypoxia inducible factor (HIF) [419], which inhibition has been shown to protect against ischemic injury in the brain [420]. Mild protein restriction was found to increase SIRT1 levels in mice [421]. Mice lacking SIRT1 exhibit altered metabolic pathways in their neurons, and a reduced capacity to mediate resveratrol-induced ischemic tolerance through glycolysis as an alternate energy source. This evidence indicates that SIRT1 is able to change metabolic pathways in neuronal cells and can confer a protective effect in emergency situations [422]. Furthermore, SIRT1 is upregulated in the substantia nigra in mice after exercise, and plays a role in reduction of inflammatory gene expression [423].

In the brain, a link between circadian clock and metabolism exists. Sleep restriction together with circadian disruption alters metabolism and increases the risk of obesity and diabetes [424]. The opposite regulation also holds true, with HFD disrupting the circadian rhythm of mice, both behaviorally and molecularly [425]. Genome-wide investigations revealed that components of the insulin signaling, the cell cycle and folate metabolism are regulated through the circadian clock [426]. SIRT1 deacetylase activity is regulated in a circadian manner [427] and counteracts the activity of the core circadian protein CLOCK, which is a histone acetyltransferase (HAT) [428]. In addition, NAD+ levels fluctuate in a circadian manner [428] and regulate transcription of stress and metabolic genes [429]. In older mice, these NAD+ transcriptional oscillations and mitochondrial respiration rhythms are restored by providing NAD+ precursors [429]. CR resets the circadian clock independent of meal timing [430]; since SIRT1 is downregulated in the brain during CR, it could play a role in managing the circadian clock. Since a robust circadian rhythm contributes to homeostasis and health, CR may be a possible way to balance the risks in people with a dysregulated circadian clocks, such as shift workers [431]. Interestingly, a timed HFD, e.g., access to food only 8 h per day instead of a constant food intake, is able to reset the circadian clock in mice and protect against obesity, when compared to mice fed with a constant food intake. This highlights the relevance of timing in a caloric sufficient diet [432]. Another way to influence the circadian rhythm is through light flicker treatment, which increases SIRT1 levels in the hypothalamus [433]. It has been shown that this therapy also normalizes hepatic lipid metabolism after ethanol exposure, suggesting a potential treatment for alcoholic liver steatosis.

In brain tissue, CR was found to reduce glucose and increase ketone bodies levels [434], as well as to increase white matter integrity and to improve long term memory in aging mice [435]. Moderate ketosis has been reported to result in multiple beneficial effects to brain function. A ketogenic diet is neuroprotective through reduction of ROS levels, by increasing mitochondrial uncoupling protein levels and activity [436], as well as by increasing oxidation of NADH and enhancing mitochondrial respiration [437]. This diet results in increased SIRT1 levels in mice hippocampal neurons [438]. Ketosis may also activate SIRT1 due to an increased NAD+/NADH ratio, which in turn can promote autophagy in the brain [439]. Ketosis-mediated increase of SIRT1 activity is supported by an increased mitochondrial biogenesis and a reduced mTOR activity, which are also reported during a ketogenic diet [440,441]. Mitochondrial SIRT3 is able to promote ketone body utilization in the brain [356]. A ketogenic diet is able to improve mitochondrial biogenesis through the SIRT3/PGC1α pathway in mice [442]. Synergy between Sirtuin activity and ketosis has been implicated in the neuroprotective effect against ischemic stroke, through the increase of NAD+ and subsequently of SIRT3 activity in the brain [443]. NAD+ increase also likely increases the activity of other Sirtuins besides SIRT3. The effect of a ketogenic diet on Alzheimer’s disease progression is currently unclear, as it has been reported that it might help [444], as well as that it does not result in any positive effect [445]. In the central nervous system, microglia are the primary source of ROS and are thought to play a role in brain aging [446].

Activation of hypothalamic FOXO1 increases food intake and body weight, and inhibits insulin and leptin signaling [447], whereas inhibition of hypothalamic FOXO1 by insulin and leptin signaling reduces food intake and thereby body weight in obese rats and increases insulin sensitivity. No similar effects were observed after blocking FOXO1 in the normal diet, non-obese control group, thus pointing to a possible therapeutic approach against clinical obesity [448]. Of note, blockage of FOXO1 was achieved by a rather invasive intracerebroventricular injection of FOXO1-antisensee oligonucleotides [448]. In mice dopaminergic neurons, FOXO1 regulates body weight, glucose and insulin homeostasis, energy expenditure and brown fat tissue [449]. Several studies investigating the role of FOXO1 in different regions of the hypothalamus have shown the overarching function of being anabolic and regulating the energy balance [450,451,452]. Heterogeneity in hypothalamic FOXO1 function is illustrated by different knockouts: a global hypothalamic knockout of FOXO1 results only in a mild decrease in body weight early in life, which normalizes over time, whereas neuronal-region specific knockouts protect against the consequences of HFD [453].

SIRT3 reduces cellular ROS levels in microglia, through deacetylating and thus activating FOXO3a, which can then upregulate antioxidant genes [454]. AKT-dependent phosphorylation of FOXO3a is reduced in neurons exposed to oligomeric Aβ; in response, to Aβ signaling, FOXO3a is activated and executes neuronal death [455]. Regulation of FOXO3a contributes to CR-induced prevention of Alzheimer’s disease and spatial memory deterioration in mouse models, and *in vitro* in human cells, by reducing Aβ peptides through insulin receptor-induced hyperphosphorylation and SIRT1-mediated deacetylation of FOXO3a [456]. SIRT3 overexpression attenuates Aβ-induced neuronal hypometabolism and improves learning and memory in mice [457], possibly through deacetylation of FOXO3a [454], as SIRT1 does. In mice brains, FOXO3a and FOXO6 are downregulated after HFD, with FOXO6 being reduced by about 80% across all brain regions [458]. This observation may suggest a role for FOXO transcription in obesity-associated cognitive impairment, as FOXO6 has been implicated in memory consolidation and synaptic function [459]. In the brain of monkeys with long standing diet-dependent obesity, it was shown that AKT and other insulin signaling factors are maximally increased, resulting in hyperphosphorylation of mTOR, FOXO1, FOXO3a, FOXO4 and subsequent decrease in PGC1α expression that, together, maintain neuronal integrity. Insulin-dependent increases in Aβ, which can promote Alzheimer’s disease, are also reported [460]. The PI3K/AKT/mTOR pathway plays a role in the inhibition of autophagy in the brain, thereby stimulating development of cognitive impairment. CR inhibits the mTOR pathway and, therefore, increases autophagy, which ameliorates age-related cognitive decline [461]. Inhibition of mTOR during CR is SIRT1-dependent [462]. Short-term fasting also inhibits mTOR, improving neuronal autophagy [463], and it may thus be used as a potential therapy against cognitive decline.

In the hypothalamus and in astrocytes, glucose, via AMPK, can inhibit oxidation and esterification of certain long-chain fatty acids [464]. This evidence may be relevant for a therapy against diabetes, as inhibition of lipid oxidation is sufficient to restore glucose and energy homeostasis in overfed rats [465]. In the presence of lipopolysaccharide, activation of AMPK maintains blood brain barrier integrity by reducing ROS [466]. Glucose, through AMPK, regulates Aβ protein levels *in vitro*. Upon AMPK inactivation in high glucose concentrations, a subsequent activation lowers Aβ production; the opposite scenario, deactivating AMPK in low glucose concentrations, was also observed [467]. Exercise was shown to activate lysosomal function in the brain through the AMPK/SIRT1/TFEB pathway, protecting against neurogenerative diseases by increased clearance of mutant proteins [468]. In addition, activation of SIRT1 and AMPK by ginsenoside Rg1 increases neurogenesis in the hippocampus of adult mice through inhibition of NF-κB [469]. However, activation of AMPK in the brain is not uniformly good, as metabolic stress can inhibit axon formation, as well as axon growth in an AMPK- and mTOR-dependent manner [470].

SIRT2 plays a role in insulin resistance in the brain. Specifically, inhibition of SIRT2 affects insulin signaling at the level of AKT, and increases insulin-stimulated glucose uptake in insulin-resistant neuronal cells [471]. Silencing of SIRT2 in a rat brain cell line reduces ATP levels [472] and increases oxidative stress and necrosis. Consequently, inhibition of anorexigenic POMC neurons and activation of AgRP neurons occur, which result in an increased feeding behavior and weight gain as seen in a HFD-fed SIRT2 knockout [473]. SIRT2 is upregulated during neuronal ischemia in oxygen-glucose deprivation, whereas its downregulation protects neurons from ischemic injury [474]. SIRT2 inhibition is also able to decrease synthesis of sterol and to exert a neuroprotective effect in cellular and rodent models of Huntington’s disease [475]. SIRT2 also interferes with autophagy-mediated degradation of protein aggregates in mouse and human cell lines, which likely explains the protective effect against Huntington’s disease [476]. Inhibition of SIRT2 *in vitro* also reduces cholesterol synthesis, which may have positive effects against the progression of both Huntington’s and Parkinson’s disease [477]. However, an *in vivo* study seems to contradict these results [478]. Increased SIRT2 levels were found in oligodendrons, which resulted in inhibition of oligodendroglial differentiation by deacetylation of the microtubule skeleton [479,480]. In neurons of the hippocampus, neurite outgrowth and growth cone collapse are inhibited by SIRT2 overexpression, most likely due to microtubule inhibition [481]. SIRT2 can be inhibited by Cyclin E/CDK2, Cyclin A/CDK2 and CDK5 phosphorylation, both *in vitro* and *in vivo* [481]. Of note, SIRT2 does not have solely negative effects on the brain, as it also plays an anti-inflammatory role in microglia and in the brain [482]. This role is supported by the fact that reduction of SIRT2 levels markedly increased inflammation markers, free radicals and thus neurotoxicity.

In mice that physically exercise on a running wheel, an increased expression of SIRT3 in the hippocampus was observed, to be essential for neuroprotective effects [483]. Phy-sical exercise also increases SIRT3 expression in the mice substantia nigra [423]. During glucose-oxygen deprivation *in vitro*, SIRT3 upregulates PGC1α and PPARγ, protecting against hypoxia by reducing ROS production and maintaining ATP levels [484]. SIRT3 deficiency-induced mitochondrial dysfunction and neuroinflammation in MetS could be causes of cognitive decline [485]. Hyperglycemia was shown to suppress SIRT3 in diabetic rats during intracerebral hemorrhages. Introducing a SIRT3 agonist ameliorates oxidative stress and mitochondrial dysfunction [486]. Administration of metformin to diabetic mice resulted in an improved mitochondrial functioning through upregulation of mitochondrial chaperones, and SIRT1, SIRT3 and PGC1α in brain tissue [487]. Metformin thus might have a neuroprotective effect in the brain of diabetic patients. Intermittent fasting can improve mood and cognition; however, improvements were not observed in SIRT3-deficient mice, indicating a critical connection between SIRT3, cognition and diet [488].

In the brain, PPARγ agonists are able to confer a neuroprotective effect, by increasing cellular respiration and PGC1α expression [489]. Glutamine metabolism, as the main excitatory neurotransmitter and precursor of GABA, is complex and highly compartmenta-lized in the brain [490]. The mTOR pathway stimulates glutamine metabolism through SIRT4 repression, which may have implications for brain signaling [491]. SIRT4 levels in the brain of the fish *Megalobrama amblycephala* were found to increase after oral glucose consumption [492]. Overexpression of SIRT4 in human glioblastoma cells was shown to prevent neuronal cell death through upregulation of glutamate metabolism [493]. Progression of fatty liver disease into cirrhosis can result in increased levels of ammonia in the brain, which potentially inhibits mitochondrial respiration. Overexpression of SIRT4 alleviated this inhibition in astrocytes by blocking glutamate metabolism in rat and human astrocytes [494].

Increased levels of SIRT5 were found during CR and linked to slower cognitive decline [495]. SIRT5 protects against metabolic and ischemic stress in the brain [496]. An increased SIRT5 expression has been observed mainly in neurons and endothelial cells within the brain, specifically located at mitochondria [497], and SIRT5 knockout mice indicated a role for SIRT5 in the glutamate-glutamine cycle and the purine metabolism.

SIRT6 is uniformly expressed throughout the entire brain, and is downregulated du-ring cerebral ischemia [498]. In the brain of *M. amblycephala*, levels of SIRT6 are upregulated after oral glucose consumption [492]. However, SIRT6 levels were also upregulated in rats during CR [499]. This evidence suggests that SIRT6 function might not be conserved across species, or that SIRT6 may have multiple roles depending on the fed state of the organism. The role of SIRT6 seems to be related to its specific location in the brain, as it induces apoptosis in cerebellar granule cells and cortical cells, but it is protective in hippocampal cells [500]. Overexpression of SIRT6 in cortical and hippocampal cells reduces cell viability during oxidative stress, but had no detrimental effects under normal conditions. Thus, SIRT6 likely requires a stimulus to induce cell death [500]. During oxidative stress, SIRT6 induces autophagy through attenuation of AKT signaling, whereas inhibition of autophagy reduces neuronal injury. Furthermore, SIRT6 inhibition suppresses autophagy and ROS induced neuronal damage [501]. Lower levels of SIRT6 are found in brains of young diabetic mice as compared to lean controls [502]. Neural SIRT6-deleted mice can reach a normal size, but ultimately become obese, indicating the critical role of SIRT6 for metabolic homeostasis [503]. SIRT6 deletion results in hyperacetylation of histones H3K9 and H3K56, suggesting that histone deacetylase activity of SIRT6 may play a role in managing obesity in rats [503]. SIRT6 deletion also resulted in reduced levels of growth hormone and hypothalamic neuropeptides that might be linked to the changes in the chromatin state. Altogether, this evidence suggests that SIRT6 may be a regulator of the onset of obesity in adult rats. No data are currently available in humans.

SIRT7 is expressed at a low level in the brain, and play a marginal role in adult neurogenesis in mice [479]. No information is currently available about a metabolic role of SIRT7 in the brain [41], and further studies are necessary to investigate this aspect.

An overview of the role of Sirtuins in the brain is presented in Table 3. An apparent heterogeneity can be observed in the functioning of Sirtuins in different regions, as well as in subsets of neurons within those regions. This heterogeneity makes it complex the design of an appropriate drug, as it is challenging to target drugs to a specific brain region. SIRT1 seems to have an overall protective effect for a number of neuronal disorders, and in most regions, also has a positive effect on metabolism. However, it has a negative effect in the hypothalamus, and this can be transmitted to other tissues. Sirtuin modulators in the brain have a greater impact than in peripheral tissues, reason of concern for drug administration that can cross the blood–brain barrier. Besides SIRT1, other Sirtuins were not extensively studied in the various regions of the brain. Nonetheless, SIRT2 seems to have an overall negative effect on the brain, whereas SIRT3 has a positive effect. SIRT6 effects are heterogeneous, but seem to be predominantly negative. SIRT6 knockout phenotypes suggest a role in metabolism. SIRT5 is reported to have a positive effect, but evidence is scarce. The role of SIRT4 in metabolism is largely unknown, however the evidence that stimulates glutamine metabolism suggests that it may be involved in other metabolic processes. A role for SIRT7 in the brain is currently unknown.

### 6.4. Adipose Tissue

In adipose-derived stem cells, the expression of SIRT1 through SIRT6 correlates ne-gatively with BMI and PPARγ expression and positively with PPARδ expression [504]. Furthermore, it has been recently shown that SIRT2 and SIRT6 in the adipose tissue were less sensitive to CR than SIRT1 and SIRT3 [505]. Moreover, FOXO1, PPARγ and adiponectin are negatively regulated in tissue samples of obese individuals [506]. When wild type adipocytes are treated with TNFα, a decrease in SIRT1 and an increase of PPARδ are observed, similarly to what is observed in cells from obese individuals. This indicates that changes in cells of obese individuals may be caused by inflammation [507]. Furthermore, SIRT1 levels in the adipose tissue correlate negatively with insulin-resistance parameters and positively with adiponectin expression [508]. SIRT1 represses adipogenesis, and its absence in mice is related to the formation of the white adipose tissue [509]. Moreover, adipose-specific knockout of SIRT1 in mice results in obesity and insulin resistance [510]. Downregulation of SIRT1 during obesity—when new adipocytes need to be constantly generated—seems therefore to be expected. However, when activating SIRT1 in mice, an increased angiogenesis with a subsequent healthy expansion of the adipose tissue occurred, resulting in an increased glucose tolerance [511]. Therefore, it is unlikely that low levels of SIRT1 during obesity are functional. SIRT1 regulates adipogenesis by interacting with, and deacetylating PPARγ, thereby inactivating PPARγ target genes. As a consequence, adipogenesis is halted [512]. Currently, no clear explanation for these conflicting results has been provided. Nonetheless, it is known that interaction between PPARγ and SIRT1 can be induced through fasting or resveratrol administration. Thus, adipogenesis may be fine-tuned by dynamic regulation of PPARγ [512].

SIRT1 levels in offspring of maternal HFD-fed mice are reduced in the adipose tissues [348], suggesting a role for SIRT1 in cross-generational metabolic diseases. CR delays development of insulin resistance, glucose intolerance, and dyslipidemia in the adipose tissue of normal mice, whereas lack of functional SIRT1 increases the incidence of those outcomes [513], indicating a role for SIRT1 against development of MetS-related symptoms. During CR, SIRT1 represses PPARγ and promotes lipolysis and loss of fat mass [248]. SIRT1 knockout in adipocytes of HFD-fed mice initially leads to glucose intolerance, hyperinsulinemia and insulin resistance faster than in wild type mice. However, upon a prolonged HFD, eventually PPARγ activity increases, leading to increased adipogenesis, improved glucose tolerance and insulin sensitivity [514]. This outcome indicates that, depending on an individual’s duration of an unhealthy diet or of obesity, different interventions could be optimal. SIRT1 decreases the level of inflammation in the adipose tissue through interaction with AKT and inhibition of the mTOR pathway in HFD-fed mice; furthermore, administration of resveratrol increases SIRT1 expression and reduces inflammation [515]. SIRT1 expression is inversely correlated with macrophage recruitment and inflammation in the adipose tissue, with SIRT1 knockdown resulting in inflammation. It has been shown that overexpression of SIRT1 in HFD-fed mice leads to a reduction in macrophage recruitment and inflammation [516], and that a moderate SIRT1 overexpression in mice protects the brown adipose tissue from inflammation, insulin resistance and defective thermogenesis [517]. The reduced function or presence of SIRT1 in HFD-fed mice or during obesity may be explained by the observation that HFD induces cleavage of SIRT1 proteins [518]. SIRT1 can improve glucose homeostasis and tolerance in mice, by enhancing the function of the brown adipose tissue; conversely, SIRT1 deficiency is accompanied by brown adipose tissue dysfunction, which is characterized by lower thermogenic activity and loss of mitochondria and mitochondrial function [519,520]. SIRT1 is able to induce a brown fat-associated phenotype in the white adipose tissue through repression of PPARγ [521].

Brown fat-associated phenotype, such as increased thermogenesis and mitochondrial synthesis in the white adipose tissue, is often referred to as beige adipose tissue. This phenotype can be induced in mice by several compounds, among which resveratrol results in the activation of thermogenesis-related genes through SIRT1/PGC1α [522]. Administration of resveratrol to activate SIRT1 also increases brown adipose tissue thermogenesis, energy expenditure and reduce fat accumulation in mice fed a standard diet [523]. Root extract from *Platycodon grandiflorus* increases browning of the white adipose tissue to create beige tissue, while increasing levels of SIRT1 and SIRT3, among others. The browning process is accompanied by lipolysis and thermogenesis in HFD-fed obese mice [524]. Coumestrol induces expansion and activation of the brown adipose tissue in HFD-fed mice, thus improving glucose tolerance and lowering body weight. Mechanistically, coumestrol increases mitochondrial content in the adipose tissue, which is associated with increased AMPK and SIRT1 levels [525]. Gallic acid improves glucose tolerance and increases thermogensis in the brown adipose tissue of HFD-fed mice [526]. Resveratrol upregulates FOXO1 and adiponectin in human adipocytes while simultaneously downregulating PPARγ, possibly through Sirtuin activation [527]. The increase in adiponectin likely results from the combined action of both SIRT1 and FOXO1 transcription, since they have been implicated in upregulating gene expression of adiponectin [528]. In addition, resveratrol increases browning of the white adipose tissue and reduces fat in both mice and human adipocytes via SIRT1 [529]. Downregulation of miR-204—which is increasingly expressed in obese individuals—also promotes browning of the white adipose tissue. This is achieved by the long-non-coding RNAs (lncRNA) that remove the inhibition on SIRT1 and PGC1α expression and AMPK phosphorylation [530,531]. Furthermore, mIR-34a levels correlate with lipid accumulation and increase hepatic steatosis through reduced fatty acid oxidation and increased mitochondrial damage by targeting SIRT1 [532]. In human obese subjects, mIR-34a also modulates SIRT1 levels [533]. Algae supplements were found to reduce weight gain and inhibited size of adipocytes, while improving serum insulin and glucose levels through upregulation of AMPK and SIRT1 [534]. Purple maize extract rich in ferulic acid and anthocyans was found to prevent HFD-induced obesity in mouse models by promoting white fat browning and thermogenesis, while reducing inflammation, partly through upregulation of SIRT1 and AMPK [338]. Another compound, allicin, regulates energy homeostasis partly through SIRT1/PGC1α; however, at higher concentrations, it leads to mitochondrial problems through SIRT5 inhibition [535]. The compound 23-epi-25-deoxyactein was found to promote adipocyte lipolysis through AMPK signaling and the SIRT1/FOXO1 pathway, and to reduce body weight gain and fat mass [536]. Moreover, 1-deoxynojirimycin, known for anti-hyperglycemic and anti-obesity effects, was found to increase glucose tolerance, insulin signaling and reduction of the white adipose tissue in HFD-fed mice [537]. This compound increases expression of genes associated with lipid metabolism such as SIRT1, FOXO1 and PPARγ and could therefore be considered for the treatment of obesity and metabolic impairment.

SIRT2 is downregulated in obese individuals [538]. Reduced SIRT1 and SIRT2 expression is an indicator of adipogenesis in human cells, whereas their overexpression reduces adipocyte differentiation [539]. Changes in SIRT1 or SIRT2 expression and adipocyte differentiation occurred in parallel with changes in PPARγ [539]. SIRT2 is upregulated in the white adipose tissue of CR mice or in cells under oxidative stress. SIRT2 deacetylates and activates FOXO3a, resulting in decreased levels of ROS [540]. A period of 24-h fasting was found to be sufficient to increase SIRT2 expression in both white and brown adipose tissue, and exposure to cold increases SIRT2 expression in brown, but not in the white adipose tissue [541]. In adipocytes, FOXO1 represses PPARγ [542]. SIRT2 decreases acetylation levels of FOXO1, which increases its binding to the PPARγ promoter. Subsequent reduction of PPARγ transcriptional activity promotes lipolysis and inhibited adipocyte differentiation [541]. During adipocyte differentiation, SIRT1 and SIRT2 are downregulated, while SIRT3 is upregulated. Overexpression of SIRT2 inhibits adipocyte differentiation, whereas a reduced SIRT2 expression results in adipogenesis. SIRT2 knockdown also increases the levels of FOXO1 acetylation, without altering SIRT1 expression levels. Thus, a reduction in SIRT2 levels is likely to play a role in adipocyte differentiation through reducing FOXO1-induced repression of PPARγ [22,541]. During hypoxia, and the resulting expression of HIF1α, a transcriptional repression of SIRT2 occurs. This observation directly translates into a reduced PGC1α deacetylation, and in a decreased expression of mitochondrial and β-oxidation genes [543].

SIRT3 is highly expressed in the brown adipose tissue, as compared to white adipose tissue, and CR or fasting increase SIRT3 expression in both white and brown adipose tissue [544,545]. A decrease in environmental temperatures upregulates SIRT3 in the brown adipose tissue, whereas higher temperatures exhibit the opposite effect [545,546]. This result occurs because SIRT3 controls brown fat tissue thermogenesis by deacetylating se-veral proteins involved in thermoregulation [547]. PGC1α induces SIRT3 expression in the white adipose tissue, resulting in cells carrying a brown adipose tissue-specific set of thermogenic gene expression. Cells lacking SIRT3 are unable to attain this gene expression set, indicating that PGC1α-mediated SIRT3 expression is required for the generation of a brown adipose differentiated phenotype [548]. Mice lacking SIRT3 showed hallmarks of fatty acid oxidation whilst fasting, including a reduction in ATP levels [546]. A forced expression of SIRT3 enhances PGC1α expression, likely through an increased phospho-rylation of CREB, which is reported to directly activate the PGC1α promoter [545,549]. An increased phosphorylation of CREB might occur through AMPK, which is activated by SIRT3 in other tissues and interacts with CREB [550]. Altogether, it appears that PGC1α and SIRT3 work together to increase their expression during certain circumstances.

An increased SIRT3 expression increases cellular respiration whilst decreasing the amount of ROS, and obesity correlates with a decreased SIRT3 expression in the brown adipose tissue of obese mice [545]. Biopsies from human tissues showed that weight loss induces SIRT3 expression [551]; conversely, obese human individuals exhibit a reduced SIRT3 expression [538]. Furthermore, reduced SIRT3 levels as well as increased levels of acetylation of proteins associated with mitochondrial biogenesis in the adipose tissue are found in obese children [552]. A SIRT3 agonist is able to induce weight loss in mice, indicating an important role for weight management through mitochondrial function [553]. The lack of SIRT3 in obese individuals may increase ROS levels, which can contribute to the inflammation of the adipose tissue. In turn, inflammation contributes to the development of MetS. Lack of SIRT3 expression is in part likely due to the decreased need for thermogenesis in obese individuals. During hypoxia in the adipose tissue, which—as described earlier—is frequently observed in obesity, SIRT3 expression is downregulated in the visceral adipose tissue. In parallel, PPARγ is upregulated, and lipogenesis is decreased. Adipose tissue explants in hypoxic conditions exhibit an altered expression of genes involved in de novo lipogenesis pathways. Lack of lipogenesis in the visceral fat could lead to an increase in circulating FFA levels [554].

SIRT3 is dispensable for maintaining brown and white adipose tissue’s mitochondrial function and whole-body metabolism. Adipocyte-specific SIRT3 knockout in mice has no major impact on putative SIRT3 targets, key metabolic pathways, and mitochondrial function in white and brown adipose tissues [555]. However, germline SIRT3 knockout mice exhibit an enhanced susceptibility to HFD-induced obesity, and display systemic metabolic complications, such as glucose intolerance, insulin resistance and NALFD [555]. Thus, the role of SIRT3 may be systemic. An alternative explanation is that local knockout in adipose tissue is compensated by another molecule, possibly another Sirtuin. Finally, depletion of SIRT3 compromises the ability of adipose-derived mesenchymal stem cells to undergo adipogenic differentiation, and leads to adipocyte dysfunction and insulin resistance [556]. In mice, treatment with schizophyllan results in an increased SIRT3 expression and activation of the brown adipose tissue, which in turn results in reduced complications in a mouse lipodystrophy model [370]. The compound increases mitochondrial metabolism and mediates mitochondrial resuscitation through SIRT3 and subsequently the downstream activation of succinate dehydrogenase A, which is deacetylated and activated in adipose tissues through mitochondrial SIRT3. SIRT3 also plays a role in adipose tissue inflammation, through its function in macrophages and ameliorating inflammatory crosstalk between macrophages and adipocytes; thus, modulation of SIRT3 may be beneficial for the treatment of adipose tissue inflammation-related metabolic disorders [557].

An HFD was shown to increase SIRT4 levels in mice [558]. SIRT4 stimulates lipoge-nesis in the white adipose tissue, whereas SIRT4 knockout mice are protected against diet-induced obesity and show an increased exercise capacity [27].

SIRT5 regulates brown adipose tissue’s differentiation, and SIRT5 knockout results in less browning and less cold tolerance [559]. Brown adipose tissue dysfunction is likely because of an increased succinylation of the mitochondrial uncoupling protein 1 (UCP1) in SIRT5 knockout [560]. Contradicting results were shown, with SIRT5 knockout in mice having no impact on the amount of brown adipose tissue and on the cold tolerance as compared to control. Epididymal white adipose tissue weighs less in SIRT5 knockout mice, and lack of SIRT5 does not protect against, or promote diet-induced obesity [561]. The latter study reported some alternate effects when mice where stressed, so that the difference of the outcome between the two SIRT5 experiments might be related to the way the mice were handled. Furthermore, a study in monozygotic twins showed that SIRT5 is downregulated in the adipose tissue of the heavier twin [562]. SIRT5 is likely downregulated during obesity, due to its role in mitochondrial respiration and β-oxidation under control of AMPK and PGC1α [373,563].

SIRT6 expression in the adipose tissue is downregulated during obesity in humans [538]. Administration of metformin to obese individuals increases SIRT6 levels in adipose tissue [564]. In mice, SIRT6 overexpression during HFD results in a significantly less accumulation of visceral fat, as well as in a reduced expression of PPARγ-responsive and lipid storage-related genes in adipocytes [565]. Activation of the AMPK and SIRT6 pathway reduces ageing-related adipose deposition induced by MetS [387]. Of note, maternal exercise of mice prior to and during pregnancy results in a reduced cardiovascular risk through upregulation of SIRT6 in heart tissue of male offspring. This in turn results in an improved fitness, defined by an increased running speed [566]. SIRT6 also reduces lipid synthesis and adipose differentiation in HFD-fed obese mice. In bovines, this same effect was reported; SIRT6 in bovines cooperates with SIRT5 to decrease lipid deposition in preadipocytes, and to inhibit adipocyte differentiation through activation of AMPK [567]. A compound, kynurenic acid, was found to reduce insulin resistance, while palmitate induced inflammation in adipocytes by increasing fatty acid oxidation through AMPK and SIRT6 in mice [568]. Inhibiting mTORC2 in brown adipocyte of mice protects against obesity at thermoneutrality. This occurs because mTORC2 inhibition triggers FOXO1 deacetylation by SIRT6, and FOXO1 drives lipid catabolism upon mTORC2 loss. This process is independent from the mTORC2/AKT pathway, and represents an interesting angle for further investigations [569]. In addition, SIRT6 plays a role in adipogenic differentiation in bovines and can be inhibited by miR-33 [570].

SIRT6-deficient mice exhibit an increased glucose uptake in the brown adipose tissue, resulting in a hypoglycemia phenotype without the normally associated hyperinsulinemia [37]. Because SIRT6 can deacetylate histone H3K9 to control expression of several glycolytic genes, SIRT6 deficient cells also exhibit an increased glycolysis and a reduced mitochondrial respiration. Thus, SIRT6 can regulate glucose homeostasis, and it might be a potential target for the treatment of T2DM.

SIRT7 expression positively correlates with obesity [571]. Upregulation of SIRT7 promotes adipogenesis through inhibition of SIRT1 [572], whereas a SIRT7 deficiency in preadipocytes reduces the ability to initiate adipogenesis [573]. Knockdown of SIRT7 increases levels of SIRT1, which in turn reduces adipogenesis and adipocyte differentiation in mice [574]. In this condition, SIRT1 binds the PPARγ promoter three times more compared to wild type, suggesting that SIRT1 may inhibit PPARγ transcription and, subsequently, adipogenesis. Thus, SIRT7 promotes adipogenesis through inhibition of SIRT1; its histone deacetylase activity is responsible for the reduction of adipogenesis [574]. A decrease of SIRT7 expression was also observed during obesity in humans [575], thus, further research is required to elucidate whether or not a difference exists across organisms in the cross-talk between SIRT1 and SIRT7.

Deletion of mTOR in mice results in a decreased mass of both brown and white adipose tissues, while also inducing the browning of the latter [576]. Moreover, ablation of mTOR in adipocytes resulted in insulin resistance and fatty liver, and it appears that mTOR is required for the differentiation of adipocytes. Furthermore, induction of PPARγ increases differentiation in adipocytes lacking mTOR [576]. Knockout of mTOR also attenuates the effects of PPARγ on lipid uptake and growth of adipocytes, resulting in hyperlipidemia [577]. Earlier studies already showed that mTOR and PI3K signaling are required for adipocyte differentiation in a PPARγ-dependent manner, and that rapamycin was able to reduce adipocyte differentiation [246,578]. It was found that AMPK suppresses mTOR signaling during adipose differentiation into brown fat, and the inhibition of AMPK halts differentiation of cells into the brown adipose tissue but leaves the white adipose tissue differentiation unaffected [579]. Resveratrol activates AMPK and induces formation of brown-like adipose phenotype in the white adipose tissue that may have a positive effect against obesity, due to an increased fatty acid oxidation in the brown-like adipose tissue [580]. The role of AMPK in the white adipose tissue has been extensively reviewed [581]. Specifically, AMPK exerts an anti-lipogenic effect through regulation of the uptake of both fatty acid and glucose, and can inhibit differentiation and proliferation in white adipocytes. FOXO3a regulates adipocyte differentiation through induction of PPARγ expression, and mice with a FOXO3a ablation exhibit a decreased amount of fat tissue due to reduced PPARγ levels [582].

An overview of the role of Sirtuins in the the adipose tissue is presented in Table 4. SIRT1 and SIRT2 have an overall positive metabolic role. SIRT2 effect is remarkable here, since it has negative effects on the liver and brain. SIRT3 expression greatly differs between brown and white adipose tissues, but it generally has a positive effect. SIRT4 generally has a negative effect, however the exact mechanisms are currently unknown. SIRT5 seems to play a role only in the white adipose tissue, which seems to be positive; however, also in this case, further research is required to investigate its specific mechanisms of action. SIRT6 seems to have a positive role, but data are still limited, and SIRT7 has an ambiguous effect. The connection between PPARγ and FOXOs is well studied only for SIRT1 and SIRT3. The AMPK/Sirtuin axis is likely to play a significant role in the adipose tissue. In general, the roles of mTOR, p53 and CDK in combination with Sirtuins are not that well established. However, these interactions are likely to be involved, especially since the adipose tissue is one of the few metabolic tissues that, during MetS, is highly connected with the cell cycle in adipogenesis.

### 6.5. Skeletal Muscle

In the skeletal muscle, CR increases SIRT1-dependent deacetylation and activity of PGC1α, which activates the genes responsible for mitochondrial fatty acid oxidation and leads to a switch from glucose to fat metabolism [583]. This metabolic switch is dependent on the AMPK/SIRT1 axis, with AMPK sensing changes in the available energy, initiating the downstream activation of SIRT1 and, subsequently, of PGC1α. AMPK also directly phosphorylates and activates PGC1α. As a result, AMPK deficiency compromises the metabolic switch in mice [584]. A study in humans indicates that a diet low in carbohydrates, but still high-caloric due to fat content, activates the AMPK/SIRT1/PGC1α pathway as if individuals adhered to CR [585]. Deacetylation of PGC1α by SIRT1 also occurs during exercise-induced mitochondrial biogenesis [586,587]. Exercise increases expression of SIRT1 in human skeletal muscle cells [586,587]. Overexpression of SIRT1 increases the proliferation rate of muscle precursor cells, thereby offsetting age-related muscle atrophy [588]. In rats, exercise increases SIRT1 expression and, subsequently, PGC1α activity; also exercise increases AMPK and FOXO3a and attenuates age-related muscle atrophy [589].

However, in mice, upregulation of SIRT1 alone is not correlated to an enhanced glucose homeostasis or to mitochondrial biogenesis as compared to wild type during exercise training [590]. This observation indicates that, even though SIRT1 is upregulated during exercise in skeletal muscle, increased levels do not mediate an enhanced response. Thus, other factors during exercise may be more important to regulate physiological response [590]. In mice, resveratrol increases mitochondrial activity and aerobic capacity in muscle cells through activation of the SIRT1/PGC1α pathway [591]. Furthermore, the drug telmisartan ameliorates insulin resistance in muscle cells by activating the AMPK/SIRT1 pathway in diabetic mice [592]. Vitamin K2 supplements in mice also increase mitochondrial functioning via SIRT1 pathways and alleviates insulin resistance in the skeletal muscle tissue [593]. Another compound, celastrol, also increases mitochondrial functioning via the AMPK-SIRT1 pathway during an HFD diet [594]. Algae extract from *Undaria pinnatifida* increases mitochondrial biogenesis, oxidative phosphorylation, oxidative muscle fiber and angiogenesis in mouse muscle tissue and, simultaneously, an increase in the activity of the AMPK/SIRT1/PGC1α pathway was observed [595]. Furthermore, mice lacking peroxiredoxin 6 (Prdx6) exhibit an early-stage DM, lower SIRT1 levels and an increased FOXO1 activity, while muscle differentiation is decreased [596]. These mice also had significant muscle atrophy compared to wild type mice.

Upregulation of SIRT1 in muscle cells protects against the effects of glucose-induced insulin resistance, through the reduction of mitochondrial dysfunction and oxidative stress, by increasing levels of SIRT3 [597]. However, other studies showed that upregulation of SIRT1 in skeletal muscle cells does not increase the total energy expenditure or enhance insulin sensitivity in mice fed a normal diet or HFD, nor does SIRT1 overexpression alters glucose-induced insulin resistance in rats [598,599,600]. In humans, SIRT1 can be regulated by fatty acid types, and can be downregulated after eating a diet with a high amount of saturated fats [601]. However, this regulation does not affect insulin sensitivity, as it occurs in mice [601]. The fact that diet composition influences SIRT1 activity could explain why a positive effect has been observed *in vitro* upon addition of glucose, but not *in vivo* in HFD.

During CR in mice, modulation of PI3K signaling by SIRT1 plays a role in enhancing the insulin sensitivity of skeletal muscle cells. Lack of SIRT1 abrogated this effect completely [602]. However, in patients with diabetes, resveratrol was able to increase total energy expenditure in the muscle through activation of the AMPK/SIRT1 axis, indicating that SIRT1 alone is perhaps not sufficient to generate an effect [603]. Upregulation of SIRT1 in mice by a compound named camptothecin also promotes lipid catabolism through AMPK/FOXO1, and reduced fat and plasma triglyceride levels *in vitro* [604]. During shor-term fasting, SIRT1 levels decrease, resulting in the wasting of muscle. FOXO1 delays skeletal muscle cell regeneration and myoblast proliferation. Inhibiting FOXO1 prevents muscle atrophy and induces hypertrophy [605,606]. Activated SIRT1 deacetylates and deactivate FOXO1 and FOXO3, and activates the PI3K/AKT pathway, promoting muscle growth and reducing muscle atrophy in mice [607]. However, after muscle damage is induced by exercise, AMPK increases, resulting in increased levels of FOXO1 and FOXO3a. This outcome limits muscle hypertrophy and inhibits myoblast differentiation in a PGC1α- dependent manner [608,609,610]. AMPK activation also reduces protein synthesis in muscle cells through the downregulation of mTOR, which further contributes to the lack of muscle cell proliferation [611]. Obesity has no effect on SIRT1 expression in human muscle cells, but hyperinsulinemia increases SIRT1 expression in muscle tissue [612].

High SIRT2 levels are found in human insulin-resistant skeletal muscle cells [613]. SIRT2 is upregulated in insulin-resistant skeletal muscle cells *in vitro* [614]. Inhibition of SIRT2 increases insulin-dependent glucose uptake and improves AKT phosphorylation in insulin-resistant cells [614]. Knockdown of SIRT2 in insulin-resistant cells ameliorates insulin sensitivity in mouse muscle cells [614]; however, knocking out SIRT2 in mice could exacerbate insulin resistance in muscle cells [473]. This different outcome may indicate a difference between systemic and local effects of SIRT2, for which the mechanisms have still to be elucidated. In addition, SIRT2 induces skeletal muscle proliferation, with pharmacological inhibition of SIRT2 resulting in side effects on muscle regeneration [615].

In mice, diet and exercise have an effect on SIRT3 expression in the muscle tissue. During fasting or CR, SIRT3 and AMPK are upregulated, while HFD reduces SIRT3 levels [550]. SIRT3 knockout mice have reduced AMPK activity and levels of PGC1α, indicating that SIRT3 works together with these proteins in exerting its effects. Indeed, exercise upregulates SIRT3 and PGC1α in mice [550]. Muscle-specific SIRT3 transgenic mice show an increased energy expenditure and lower respiratory rates, indicating a shift to lipid oxidation [616]. Transgenic mice were also able to run 45% longer than control mice, and have an increased expression of AMPK and PPARδ in the muscle tissue. However, transgenic mice exhibit 30% less muscle mass resulting from a permanent upregulation of FOXO1 [616].

The expression of SIRT3 in skeletal muscle cells is reduced during diabetes, and SIRT3 knockout mice exhibit increased oxidative stress, reduced oxygen consumption and impaired insulin signaling in their muscle cells [617]. Muscle tissue-specific SIRT3 knockout mice show increased mitochondrial acetylation rates but an unaltered metabolic homeostasis and oxidative stress in either normal or high fat diets [618]. This observation indicates that systemic and muscle-specific SIRT3 knockout have different effects, potentially through an impaired brain signaling in the systemic knockout model. SIRT3 seems to have an effect on mitochondrial substrate choice and metabolic flexibility; however, inhibition of SIRT3 results in a switch from carbohydrate to lactate and fatty acids as fuel, even in a fed state [619]. This outcome is in agreement with data that indicated that SIRT3 flux, together with acetyl-lysine turnover, is responsible for promoting the fuel switch between fat and glucose [620]. Overexpression of SIRT3 in muscle cells increases glucose uptake and decreases ROS production. Moreover, metformin ameliorates insulin resistance in muscle cells by upregulating glucose uptake through SIRT3 [621]. In overweight individuals, aerobic exercise upregulates SIRT3 and PGC1α in skeletal muscle cells without changing their caloric intake. The same holds true for interval training in healthy individuals [622,623,624]. Athletes have higher SIRT1, SIRT3 and FOXO1 levels as compared to age-matched controls, indicating that exercise in healthy people can result in better cellular metabolism as well [625]. SIRT3 was found to scale with muscle oxidative capacity and mitochondrial fatty acid oxidation, and upregulation of SIRT3 after chronic muscle stimulation is AMPK-dependent [626]. In mice, SIRT3 plays a crucial role in maintaining the skeletal muscle insulin action, as well as in protecting against the effects of insulin resistance during HFD, through facilitation of glucose disposal and mitochondrial function [627]. SIRT3 knockout in mice shows that SIRT3 has an effect on autophagy. The compound dihydromyricetin activates the autophagy pathway through the AMPK/SIRT3/PGC1α axis, to improve insulin sensitivity [628]. Finally, metformin ameliorates insulin resistance in skeletal muscle cells, in part through upregulation of SIRT3, as previously mentioned [621].

SIRT4 is upregulated in muscle cells of *M. amblycephala* after a high oral glucose intake, indicating that SIRT4 plays a role in muscle cell metabolism [492]. In mice muscle cells, SIRT4 regulates ATP levels, with SIRT4 knockouts having significantly reduced ATP levels in the muscle tissue in both fed and starved conditions. Overexpression of SIRT4 results in increased ATP/ADP values, whereas the opposite occurs for SIRT4 knockouts. SIRT4 regulates ATP levels through PGC1α and AMPK [629]. At high altitudes, SIRT4 mediates the switch from fat to carbohydrate metabolism, the latter being significantly upregulated at higher altitudes in human muscle cells [630]. Furthermore, knockdown of SIRT4 increases fatty acid oxidation and cellular respiration rates of skeletal muscle cells in mice as well as AMPK expression [28]. Upon SIRT4 knockdown, SIRT1 and SIRT3 are upregulated. In addition, upon SIRT1 knockout, the increase in fatty acid oxidation halts, indicating that SIRT4 might work upstream of AMPK in the AMPK/SIRT1 pathway [28].

In the skeletal muscle of SIRT5 knockout mice, a number of proteins are hypersuccinylated; however, no major physiological effect is observed [561]. Moreover, deletion of SIRT5 does not result in changes in expression of genes involved in the regulation of metabolism in muscle cells [561]. SIRT5 levels in muscle tissue can be increased by feeding mice either a whey protein or milk protein supplement [631].

SIRT6 is upregulated in muscle cells of *M. amblycephala* after a high oral glucose intake, indicating that SIRT6 plays a role in muscle cell metabolism [492]. Moderate physiological overexpression of SIRT6 in skeletal muscle cells increases insulin sensitivity in mice [389], and upregulates physical activity and AMPK levels in mice muscle cells [632]. SIRT6 levels can be increased by feeding mice milk protein supplements, whereas exercise was shown to decrease their SIRT6 levels [631]. However, other studies showed that SIRT6 deficiency in skeletal muscle cells results in an upregulation of AKT, an enhanced recruitment of glucose transporters to the cell membrane as well as an increased insulin signaling upstream of AKT. In turn, an increased glucose uptake subsequently causes hypoglycaemia [633]. SIRT6 deficiency in muscle cells also results in impaired insulin sensitivity, weakening of exercise performance and attenuation of body energy expenditure [634]. Currently, no explanation for these inconsistent findings has been provided. However, it is suggested that a different tissue distribution of SIRT6 expression may exist [389].

SIRT6 knockout mice show a degenerated skeletal muscle phenotype accompanied by significant fibrosis, indicating that SIRT6 is vital in maintaining muscle mass [635]. SIRT6 is recruited to histone H3k9 at NF-kB binding sites, and their deacetylation results in the termination of NF-kB signaling leading to a reduced expression of atrophic factors. Thus, SIRT6 is responsible for maintaining skeleton muscle mass through its deacetylase activity [635]. SIRT6 deletion results in a decreased expression of genes involved in metabolism, through a reduced AMPK activity [634]. Conversely, overexpression of SIRT6 in muscle cells activates AMPK [634]. SIRT6 in the muscle mediates glucose metabolism and exercise capacity through activating AMPK, and can therefore be a potential therapeutic target against T2DM [634]. In older rats, SIRT6 is upregulated, and exercise attenuates its levels. Increased SIRT6 levels may cause the opposite effect of SIRT6 deficiency, namely hyperglycemia, increasing the risk of diabetes with age as levels of SIRT6 and blood glucose rise [636].

Aging decreases SIRT7 expression in rat skeletal muscle cells. Since SIRT7 regulates the muscle protein degradation system in old animals, SIRT7 may act against sarcopenia, which is characterized by loss of skeletal muscle mass and function. A syndrome linking obesity to sarcopenia, called sarcopenic obesity or obese sarcopenia, exists, which is however clinically poorly defined [637,638]. Short-term CR decreases SIRT7 levels, which remain reduced after refeeding [639]. In contrast, SIRT7 levels can be increased by feeding mice whey protein or milk protein supplements [631].

Leptin regulates muscle growth and development by acting on SIRT1, FOXO3a and their downstream targets, such as PGC1α. Activating the SIRT1/PGC1α axis protects the diabetic heart by reducing hypertrophy and fibrosis [640]. In high glucose media, leptin is associated to AKT and AMPK activation in muscle cells, which affect PGC1α post-tran-slational modifications that are reversible upon removal of leptin [641,642]. Moreover, p53 increases muscle aerobic capacity through increasing mitochondrial content. When p53 is knocked out, reduced levels of PPARγ and PGC1α are observed, and reduced apoptosis and mitochondria are observed [643,644].

An overview of the role of Sirtuins in the muscle is presented in Table 5. SIRT1 and SIRT3 have a positive effect on metabolism. SIRT2 is important for muscle proliferation, but has a negative effect on MetS progression. SIRT4 has an ambiguous effect, increasing ATP levels, but reducing fat oxidation. The role of SIRT4 in muscles during metabolism and disease required further research. SIRT5 plays no important role in muscle cells, whereas SIRT6 has an overall negative effect. SIRT7 protects against age-related degradation, however no information about its metabolic effects in muscle is currently available. Relations between most Sirtuins with AMPK and with FOXOs are documented. However, the role of Sirtuins together with mTOR, p53 and CDK is not known, although a connection with cell cycle proteins is suspected during muscle atrophy.

### 6.6. Heart

Activation of SIRT1 in cardiomyocytes prevents the development of cardiac hypertrophy, and protects the cells from inflammation and metabolic dysregulation. These effects are mediated through the interaction with PPARα and deacetylation of PGC1α, which also results in the transcription of genes involved in fatty acid oxidation [645]. CR can mediate diabetic cardiomyopathy through the SIRT1/PGC1α pathway, by decreasing oxidative stress and inflammation [646]. Eight weeks of CR and resistance exercise in HFD-fed obese rats increase SIRT1/FOXO1 expression in skeletal muscle and reduce mTOR, which—when active—is a hypertrophic factor. Yet, no difference in heart weight was found [647]. CR can also improve cardiac function in rats, through SIRT1 and SIRT3 upregulation [648]. However, suppression of the estrogen-related receptor transcriptional pathway by PPARα/SIRT1 as a physiological fasting response is involved in the progression of heart failure by inducing mitochondrial dysfunction [649]. This observation indicates that the PPARα/SIRT1 pathway can mediate the physiological adaptation of the heart during starvation, and that its inappropriate activation may lead to heart failure. PPARδ increases the capacity for myocardial glucose utilization and has a cardioprotective effect. Moreover, the ratio between PPARα and PPARδ was skewed in the diabetic heart, with less PPARδ shifting metabolism from glucose to fat [650]. Dual activation of PPARα and PPARγ leads to cardiac dysfunction associated with PGC1α suppression, and with a decreased mitochondrial abundance, likely due to competition between the two transcription factors [651].

Oxidative stress plays a relevant role in the development of a number of cardiovascular diseases [652]. SIRT1 manages the oxidative stress response through the regulation of p53, FOXO1, FOXO3 and FOXO4 [653,654,655,656]. AMPK is downregulated in diabetic patients and, as such, the failing heart is unable to meet the normal metabolic needs of the body. Therefore, AMPK plays an important role in development in heart failure, and upregulation of AMPK has been associated with protection against heart injury [657]. Therapy with resveratrol increases AMPK, PPARγ and PGC1α activities, whereas it decreases PPARα and p53 activities through SIRT1. This outcome has positive effects on heart failure, suggesting the resveratrol is promising against heart disease [658]. However, high concentrations of resveratrol can inhibit the PI3K/AKT/mTOR pathway, thus indicating that an understanding of the specific mechanisms of action of this compound is critical before to consider it a safe intervention [659]. Physiological hypertrophy is correlated to an increased cardiac angiogenesis, while pathological hypertrophy is correlated with reduced in capillary density [660]. AKT is a major player in cardiac angiogenesis, and acute AKT expression results in mTOR-dependent induction of myocardial vascular endothelial growth factors [661]. AKT is also involved in cardiomyocyte growth through mTOR [661]. However, if AKT is chronically expressed, the amount of angiogenesis promoters decreases, while hypertrophy continues and leads to cardiovascular pathologies [661]. Therefore, the balance between hypertrophy and angiogenesis is determinant for physiological and pathological hypertrophy. SIRT1 is able to activate AKT, which has two potential downstream effects: it can promote physiological cardiac hypertrophy, or it can antagonize pathological cardiac hypertrophy [662]. These results indicate that different molecular signaling pathways are active in different types of hypertrophy. Targeting specifically pathological hypertrophy would be an interesting intervention. It was found that inhibition of mTOR by rapamycin improves cardiac function in diabetic mice [663]. Mice lacking SIRT1 have a reduced physiological cardiac hypertrophy after exercise, indicating the importance of the AKT/SIRT1 pathway for a healthy heart [664]. Mice in which development of cardiac hypertrophy and heart failure was stimulated, showed an improved the energy metabolism upon administration of a compound named vildagliptin. Vildagliptin increased fibroblast growth-factor 21 (FG21) levels through SIRT1-mediated pathways and both mice and human cardiac fibroblasts [665]. FG21 stimulates PGC1α and AMPK/SIRT1 [666], and is stimulated by PPARα [665]. These interactions could result in a protective effect for cardiac energy metabolism, through increased fatty acid and glucose metabolism. AKT activation can also have a cardio protective effect after ischemic injury, by protecting cardiomyocytes from apoptosis and improving surviving cardiomyocyte function [667]. SIRT1 can promote autophagy and inhibit apoptosis after hypoxia in the heart via AMPK, thus a role for the AMPK/mTOR/AKT pathway is likely [668]. In addition, SIRT1 can also protect cardiomyocytes against apoptosis induced by stress of the endoplasmic reticulum (ER). It does this through PERK/eIF2α-, ATF6/CHOP- and IRE1α/JNK-mediated pathways [669]. Finally, SIRT1 seems to play a role in mice recovering from myocardial infarction by low-to-moderate intensity exercise [670]. The compound taurine was found to influence the cardiac tissue. It has a cardio-protective effect during pressure overload-induced heart failure by activating the SIRT1/p53 pathway, thus reducing myocyte hypertrophy and fibrosis [671]. Moreover, bioactive peptides were found to attenuate cardiac apoptosis through the activation of mitochondrial biogenesis pathways and of SIRT1 and PGC1α [672]. These bioactive peptides could be considered a future alternative anti-hypertensive drug.

In the heart, stability of microtubules plays a role in the development of cardiac di-sease, since microtubules that are too stable impair contraction of the heart [673]. SIRT2 plays a role in the stability of microtubules by deacetylation, thus making them more unstable, which potentially protects the heart [674]. In the diabetic heart, SIRT2 is downre-gulated, resulting in an increased microtubule activity and an increased risk of cardiac disease [674]. SIRT2 levels are decreased in mice during pathological cardiac hypertrophy and SIRT2 deficient mice develop pathological cardiac hypertrophy and cardiac dysfun-ction. In addition, SIRT2 promotes AMPK activation and represses nuclear factor of activated T-cells (NFAT) transcription factors. Thus, SIRT2 blocks both specific protein synthesis and gene transcription required for cardiomyocyte hypertrophy [675,676].

SIRT3 is downregulated in the hearts of obese humans and rats as well as in HFD-fed mice, and a decreased SIRT3 activity can result in mitochondrial hyperacetylation and subsequent ventricular dysfunction and heart failure [677,678]. Hyperacetylation of mitochondrial proteins can be critical in the pathogenesis of cardiac disease in mice, and it is coupled to lower SIRT3 and NAD+ levels observed in the heart during stress. Administration of the drug nicotinamide mononucleotide preserves cardiac mitochondrial homeostasis and heart failure in stress-sensitive mice [679]. SIRT3 has an anti-hypertrophic effect on heart cells through activation of AMPK and nuclear localization of FOXO3a. SIRT3 also blocks AKT activity through FOXO-mediated ROS level reduction, and increased ROS levels activate AKT. The decrease of AKT activity can be—depending on the cause of hypertrophy—either beneficial or detrimental [680]. However, FOXO signaling has been also reported to activate AKT by inhibiting protein phosphatases, and to play a role in insulin sensitivity and impaired glucose metabolism in the heart [681]. Downregulation of SIRT3 has been associated with impaired mitochondrial and contractile function of the heart [682]. Furthermore, SIRT3 inhibits FOS transcription through deacetylation of the histone H3K27 at the FOS promoter [683]. SIRT3 knockout mice exhibit an increased inflammation and fibrosis in the heart, whereas consistent overexpression of SIRT3 in mice cardiomyocytes can partially prevent inflammation and fibrosis [683]. Considering that FOS is upregulated during cardiac hypertrophy and heart failure, SIRT3 may have a cardioprotective effect through its histone deacetylase activity in the heart [683]. In rats with ischemic injury, overexpression of SIRT3 ameliorates hypoxia-induced dysfunction [684]. SIRT3 has an anti-hypertrophic effect on cardiomyocytes, through activation of AMPK and nuclear localization of FOXO3a [685]. Moreover, SIRT3 blocks AKT activity through FOXO-mediated ROS level reduction, and SIRT3 affects hypertrophy through regulation of autophagy [685]. However, the molecular mechanisms of action are currently unknown. SIRT3 levels are reduced during HFD, since the response to such a diet increases cardiac fatty acid oxidation, which is controlled through SIRT3 downregulation and subsequently increased in the acetylation of enzymes involved in mitochondrial β-oxidation [686]. An increase of SIRT3 levels can preserve heart function and heart capillary density during obesity, and activation of the SIRT3/FOXO3 pathway can protect the heart from HFD-induced oxidative stress [687,688]. Resveratrol preserves the mitochondrial function and cellular size in the heart of diabetic rats [689]. Furthermore, choline, an essential nutrient, ameliorates cardiac hypertrophy by regulating metabolic remodeling through the AMPK/SIRT3 pathway [690]. AMPK may inhibit hypertrophy and ameliorates cardiac dysfunction caused by oxidative stress in mice hearts, through the activation of SIRT3 [691]. Through SIRT3 signaling, melatonin can modulate autophagy, decrease apoptosis and alleviate mitochondrial dysfunction during diabetic cardiomyopathy in mice, and SIRT3 knockout abolishes the effect of melatonin on cardiac function [692]. Of note, the compound licoisoflavone A reduces cardiac hypertrophy by upregulation of SIRT3 [693].

SIRT4 is highly expressed in the heart and inhibits the activity of manganese superoxide dismutase (MnSOD), which manages ROS levels. Specifically, SIRT4 activity increases ROS levels and induces hypertrophic growth, generation of fibrosis and cardiac dysfunction [694]. However, SIRT4 also plays a role in the prevention of hypoxia-induced apoptosis during ischemic heart injury [695]. An increase of SIRT4 levels results in an improved protection of heart cells from ischemia through the protection of mitochondria and the reduction of apoptosis [696].

Succinyl-CoA is the most abundant acyl-CoA molecule, and a number of cardiac proteins in the heart are regulated by SIRT5, including those involved in fatty acid oxidation [697]. SIRT5 knockout mice have increased long-chain acyl-CoAs and decreased ATP levels in the heart under fasting conditions, and develop a hypertrophic cardiomyopathy [697]. Moreover, SIRT5 knockout in mice increases protein succinylation. Other Sirtuin functions, such as acetylation, where unaffected, indicating that desuccinylation is the main function of SIRT5 in the heart [698]. As a result of SIRT5 deficiency, proteins affected by an increased succinylation are mainly active in the oxidative metabolism and oxidative phosphorylation [698]. Moreover, SIRT5 knockout mice have an increased hypertrophy and a greater systolic impairment as compared to wild type [698]. It was discovered that regulation of SIRT5 occurs via upstream proteins and metabolites that inhibit a damage cascade via activation of several cytokines. Therefore, SIRT5 may represent an interesting therapeutic target against ischemia-reperfusion injury [699].

SIRT6 protects cardiomyocytes against ischemia and reperfusion injury. It does this through the upregulation of AMP/ATP and activation of AMPK, which decreases cellular levels of oxidative stress through FOXO3a [700]. SIRT6 is downregulated in HFD-fed mice. Overexpression of SIRT6 protects mice from developing obesity and insulin resistance [565]. SIRT6 and SIRT3 maintain each other’s activity and protect the heart from developing a diabetic cardiomyopathy [678]. SIRT6 binds to and suppresses the promoter of IGF signaling-related genes through interaction with the transcription factor c-Jun and deacetylating the histone H3K9 at IGF signaling-related promoter genes. Thus, SIRT6 inhibits the IGF/AKT pathway, which reduces the development of cardiac hypertrophy. In this way, SIRT6 protects the heart, and a decreased SIRT6 expression is observed in failing hearts, indicating that disrupted histone deacetylase activity of SIRT6 can lead to cardiac disease [701]. SIRT6 also possibly regulates the nuclear retention of FOXO3 through the AKT pathway [702]. SIRT6 suppressed cardiomyocyte hypertrophy *in vitro* via inhibition of the NF-κB-dependent transcriptional activity [703]. Finally, SIRT6 knockdown in mice results in a decreased oxygen consumption and ATP production, and in an increased nuclear localization of FOXO1. Subsequent transcriptional changes can cause reduced heart functioning [704].

SIRT7-deficient mice develop cardiac hypertrophy, inflammatory cardiomyopathy and fibrosis. SIRT7 interacts with and deacetylate p53, and lack of SIRT7 resulted in p53 hyperacetylation and activity [640]. Cardiomyocytes lacking SIRT7 showed an increased apoptosis and a diminished resistance to oxidative and genotoxic stress, whereas SIRT7 overexpression reduces apoptosis during hypoxia, indicating that SIRT7 is a stress survival gene in cardiomyocytes [641,642]. SIRT7 is also involved in myocardial tissue repair through the growth factor B pathway [705]. Short-term CR increases expression of SIRT1-4 and SIRT7 in cardiomyocytes, which demonstrates the beneficial effect that diet can have on cardiovascular disease progression [706]. PPARγ can have a beneficial effect on some tissues. However, the cardiomyocyte expression of PPARγ leads to cardiac dysfunction in mice [707]. The CDK inhibitors p21 and p27 are downregulated, whereas p57 is upre-gulated, during human heart failure [708]. These levels are reminiscent of the expression of the fetal heart, possibly because of an unsuccessful attempt to initiate cell division in the terminally differentiated cells [708]. Overexpression of the CDK inhibitor p16 through an adenovirus vector was shown to inhibit cardiac hypertrophy *in vivo*, suggesting a novel therapy for pathological cardiac hypertrophy [709].

An overview of the role of Sirtuins in the heart is presented in Table 6. SIRT1 and SIRT3 protect against cardiac pathologies. SIRT6 and SIRT7 also have a positive role on cardiac pathologies, although their role is not yet well described. This holds also for SIRT2, of which the metabolic functioning in the heart is not yet understood. Considering its roles in some peripheral tissues, SIRT2 represents an interesting target for further research. SIRT4′s role in cardiac disease progression is dependent on the specific pathology, and its role during hypoxia has still to be understood. SIRT5 has a positive effect in the heart, however none of the underlying mechanism properly described. The connection between Sirtuins and their main partners in the heart has been described for SIRT1 and, partially, also for SIRT3, but is lacking for other Sirtuins. Considering that cardiac hypertrophy plays a relevant role in cardiac pathologies, it can be suspected that cell cycle players such as p53, mTOR and CDKs can be involved in disease progression. Interaction of these proteins with Sirtuins in the heart should be investigated.

## 7. Discussion

Sirtuins are key players in metabolic regulation in several organisms, including humans. Thus, Sirtuins and their interactors may be implicated in molecular mechanisms underlying metabolic disorders. The most prominent interaction partners of Sirtuins are signaling regulators, such as AMPK, FOXOs, PPARs, PI3K, AKT and mTOR, and cell cycle proteins, such as cyclins and CDKs. Interactions of these molecules with Sirtuins exert tissue-dependent effects, and can either stimulate or inhibit development and progression of metabolic disorders. In Table 7 are summarized, for each Sirtuin, the tissue(s)/organ(s) where they are located, dietary and medicinal influences, and their involvement in specific metabolic disorders. Furthermore, in Figure 1, the relevant Sirtuins and their main target genes/interactions in the various tissues have been grouped according to the metabolic processes that they are involved with. Below, we discuss the interplay between specific Sirtuins to regulate the metabolic tissues at local and systemic levels.

### 7.1. Pancreas

In the pancreas, SIRT1 is the most extensively studied among the Sirtuins, and FOXO1 is its relevant interaction partner. Deacetylation of FOXO1 by SIRT1 and reversal of nuclear localization increases insulin secretion and protects pancreatic cells against stress-related apoptosis [266,267,268]. In fact, activation of SIRT1 promotes cellular repair [270]. Hyperlipidemia decreases SIRT1 expression, resulting in opposite effects [266]. GLP-1 also inhibits SIRT1 activity, leading to an increased FOXO1 acetylation, which results in an increased β-cell proliferation [271]. Interestingly, overexpression of SIRT1 results in a similar effect [273]. The main role of SIRT1 in the pancreas involves regulation of insulin secretion. With age, glucose-stimulated insulin secretion may decline, possibly due to decreased NAD+ synthesis [277]. Administration of GABA increases NAD+ levels and reduces pancreatic apoptosis in a SIRT1-dependent manner [283]. This and other strategies to increase NAD+ levels may be investigated as potential treatment. In the pancreas, other Sirtuins have been studied less extensively, and studies report contradicting results that should be further addressed to elucidate the underlying mechanisms of action.

In this tissue, each Sirtuin has specific functions; however, some Sirtuins are involved in similar metabolic pathways. For example, SIRT1 and SIRT6 play a role in glucose tole-rance [276,279,280,293,296,297] and share FOXO1 as a common target [266,267,268,270,295]. In addition, SIRT1 and SIRT6, together with SIRT3 and SIRT5, are also involved in insulin signaling and secretion [263,269,279,289,290,293,296,297]. Given the involvement in these metabolic pathways, it is not surprising that SIRT1, SIRT3 and SIRT5 were found to play an important role in T2DM [274,293,294]. A direct link between SIRT6 involvement in metabolic pathways and T2DM has been, to the best of our knowledge, not reported. Besides glucose metabolism, Sirtuins in the pancreas—specifically SIRT1 and SIRT5—also play a role in β-cell proliferation and maintenance [273,275,281]. Finally, SIRT1 and SIRT3 are involved in oxidative stress management: SIRT1 protects, through activation of FOXO1, cells against hyperglycemia-induced oxidative stress, whereas SIRT3 protects pancreatic cells in mice against palmitate-induced stress [272,288].

### 7.2. Liver

In the liver, most information is available again for SIRT1, which acts via NAD+ as nutrient sensor to regulate glucose homeostasis. SIRT1 deacetylates FOXO1 [307], which subsequently interacts with PGC1α [306]. PPARγ is also likely involved in the SIRT1-dependent regulation of glucose homeostasis [301]. In addition, SIRT1 activates AMPK and PPARα to decrease fatty acid synthesis and stimulate oxidation thereof, thus playing a role in liver lipid metabolism [308,309]. Adiponectin is an important player in the interaction between SIRT1, AMPK and PPARα [313,314], and AMPK regulates the PI3K/AKT/mTOR pathway, which acts through FOXOs to regulate in lipogenesis [315,316]. Reduction of SIRT1 expression impairs signaling in this pathway [305], as well as PPARα signaling [309]. SIRT1 levels are lower in humans with NALFD [321], as well as in HFD-fed mice suffering from NAFLD. Interestingly, an HFD increases SIRT1 levels in healthy livers, as does adherence to CR [323,324,325,326]. SIRT2 has an overall positive effect in the liver, because it suppresses hepatic fibrosis [351] and steatosis [352], and it counteracts ROS generation and mitochondrial dysfunction to restore insulin sensitivity [353]. Moreover, SIRT2 promotes hepatic glucose uptake and tolerance in mice [354]. SIRT3 deacetylates proteins in diverse metabolic pathways [355], such as the ketogenic pathways during fasting [356] and the fatty acid oxidation and the urea cycle during CR [357]. SIRT3 is also involved in mitochondrial maintenance, which is disrupted upon SIRT3 downre-gulation, resulting in lipotoxicity upon nutrient excess [358] and less ROS scavenging [363]. Anti-ROS activity results from SIRT3 deacetylating PPARγ and FOXO1 [363,365]. SIRT3 expression can be reduced by palmitic acid, which levels increase during obesity, lea-ding to oxidative stress and apoptosis [351], and to impaired intestinal permeability [366]. Conversely, SIRT3 expression increases during fasting [372]. Interaction partners of SIRT3 are AMPK, PGC1α, PPARα and CPS1. AMPK activation by SIRT3 suppresses lipogenesis and protects against lipotoxicity; PGC1α interacts with SIRT3 and CPS1 to regulate the urea cycle [374]. Contrarily to SIRT1 and SIRT3, SIRT4 exerts a negative effect in the liver. SIRT4 interacts with SIRT1 and PPARα, inhibiting their expression. By suppressing SIRT1, SIRT4 inhibits fatty acid oxidation and mitochondrial gene expression in the liver [376]. SIRT5 regulates ketogenesis and β-oxidation [31]. PGC1α and PPARα upregulate SIRT5 levels during fasting, while AMPK downregulates SIRT5 levels in this condition [373]. SIRT6 negatively regulates glycolysis, triglyceride synthesis and fat metabolism [381], which generates a protective effect for the liver [384]. Overexpression of SIRT6 protects against disrupted glucose homeostasis and increases insulin sensitivity [390]. SIRT6 activates PPARα to promote fatty acid β-oxidation, thus reducing liver fat content [392]. SIRT6 is regulated by SIRT1 and FOXO3a [382]. The interaction between FOXO3a and SIRT6 is also involved in lowering LDL levels [389]. SIRT7 positively regulates TR4/TAKI that is involved in fatty acid uptake and triglyceride synthesis. Due to studies reporting contradicting results, the role of SIRT7 in NALFD development and progression remains the subject of debates [393,394].

In the liver, Sirtuins influence glucose metabolism. SIRT1 increases gluconeogenesis [299,300,307] and decreases glycolysis similarly to SIRT6 [299,300,381]. SIRT2 stimulates glucose uptake and tolerance [354], and it increases insulin sensitivity similarly to SIRT1 and SIRT6 [305,318,337,353,389]. Lipid metabolism is also influenced by Sirtuins present in the liver. For example, triglyceride synthesis is decreased by SIRT6 and SIRT7 [381,392]. SIRT6, together with SIRT1, SIRT3 and SIRT5, also influence β-oxidation of fatty acids through the common target PPARα [30,318,324,335,357,360,390]. Contrarily, SIRT4 inhi-bits fatty acid oxidation by repressing PPARα [28,376]. Furthermore, SIRT4 reduces mitochondrial gene expression, whereas SIRT3 is involved in maintaining integrity of mitochondria through protection from ischemic damage [474,483,484,496].

### 7.3. Brain

In the brain, SIRT1 is the most extensively studied among the Sirtuins, for which a vast number of functions has been discovered, both related to metabolism and other types of functions, such as in memory [407,408] and motor function [412]. SIRT1 positively regulates FOXO1, which subsequently stimulates excessive food intake [398,399,402]. Food intake is also stimulated via the SIRT1/p53 axis that activates hypothalamic AMPK [413]. Negative regulation of FOXO1 is exerted by the PI3K/AKT pathway [398]. FOXO1 seems to have different functions in different hypothalamic regions, although its overall function is anabolic in nature [450,451,452,453]. Contrarily to its effect in peripheral tissues, SIRT1 thus exerts a negative metabolic effect in the brain, specifically in the hypothalamus. Therefore, it is important to consider adverse effects before the administration of systemic drugs that influence SIRT1 function. SIRT1 also plays a role in autophagy [414], Alzheimer’s disease [416,417], ischemic injury [418,419,420,422,431] and inflammatory gene expression [423]. SIRT1 stimulates autophagy while protecting against or inhibiting the latter three diseases under specified circumstances. In addition to SIRT1, the PI3K/AKT/mTOR pathway is also involved in autophagy, which inhibits it in a SIRT1-dependent manner [461,462]. SIRT1 affects progression of Alzheimer’s disease through interactions with FOXO3a and SIRT3 [453,456]. Activity of SIRT1 is modulated through processes influencing the NAD+/NADH ratio, with increased ratios activating SIRT1 [439]. Given that NAD+ levels fluctuate in a circadian manner [428], SIRT1 activity follows a similar pattern [427]. The major function of SIRT2 in the brain is to regulate insulin resistance by interacting with AKT [471]. In addition, SIRT2 affects ATP levels, oxidative stress and necrosis [472]. Downregulation of SIRT2 has both negative effects, by increasing oxidative stress and necrosis [472], and positive effects, by protecting from ischemic injury [474] and inhibiting progression of Huntington’s and Parkinson’s diseases [475,477]. Of note, recent studies contradict these findings, and further experiments are needed to validate the findings. SIRT3 is essential for the neuroprotective effects of physical exercise [483], by promoting an increase in PGC1α and PPARγ levels [484]. SIRT3 protects against cognitive decline, mitochondrial dysfunction and oxidative stress [485,486], and improves mood and cognition [488]. SIRT4 is thought to inhibit metabolism as its inhibition by the mTOR pathway stimulates metabolism [491]. SIRT6 seems to have a location-dependent function in the brain [500]. SIRT6 interacts with AKT to induce autophagy during oxidative stress, which is inhibited upon repression of SIRT6. SIRT6 likely plays a role in metabolism, as SIRT6 deletion ultimately results in obesity; however, the underlying mechanism of its function has to be elucidated yet [503].

In the brain, the effects of Sirtuins on local metabolic pathways is less extensive compared to the pancreas and liver, and shows less overlap among Sirtuins. However, SIRT1 and SIRT2 induce insulin sensitivity through the common target AKT [405,471]. Furthermore, in this tissue, (protection from) ischemia involves multiple Sirtuins. Downregulation of SIRT2 prevents neuronal ischemic injury, whereas upregulation of SIRT3 and SIRT5 serves a protective function [474,483,484,496].

### 7.4. Adipose Tissue

In the adipose tissue, SIRT1 and SIRT3 are the most studied among the Sirtuins. SIRT1 is a crucial repressor of adipogenesis [509], and it regulates insulin sensitivity and glucose tolerance [510,511]. Regulation of adipogenesis occurs through interaction with PPARγ [249,512]; regulation of glucose homeostasis occurs by enhancing brown adipose tissue, also through PPARγ [519,520,521]. SIRT1 is also involved in cross-generational metabolic diseases [351] and protection against MetS-related symptoms [513]. In addition, by interacting with AKT and mTOR, SIRT1 decreases inflammation in the adipose tissue [515]. Both SIRT1 and SIRT2 interact with PPARγ to regulate adipocyte differentiation [539]. SIRT2 also interacts with FOXO3a, to decrease ROS levels [540], and with FOXO1, to repress PPARγ promotors [23,541,542], which in turn promotes lipolysis and inhibits adipocyte differentiation [541]. While SIRT1 and SIRT2 are downregulated during adipocyte differentiation, SIRT3 is upregulated. SIRT3 controls brown fat thermogenesis [547], and its expression is regulated by PGC1α [548] that is activated by phosphorylated CREB [545,549]. Furthermore, SIRT3 increases cellular respiration whilst decreasing ROS levels [546]. Generally, SIRT3 is thought to have a systemic rather than a local role [555]. SIRT6 is involved in lipid storage and adipose deposition induced by MetS through AMPK, and it is mainly investigated in mice and bovines [387,565,567]. SIRT7 seems to be involved in obesity by promoting adipogenesis through the inhibition of SIRT1 [571,572]; however, conflicting findings are reported [575].

In the adipose tissue, multiple Sirtuins play a role in metabolic pathways, including adipogenesis. SIRT1 and SIRT2 inhibit PPARγ, which in turn represses adipocyte differentiation [509,512,521,539,541]. PPARγ is also downregulated by SIRT3, which results in increased lipogenesis. Through a different pathway—activation of AMPK—SIRT6 inhibits adipocyte differentiation [567,570]. Contrarily, SIRT5 stimulates adipose differentiation similarly to SIRT7 [559,572], the latter of which achieves this role through downregulation of SIRT1 [572]. SIRT1 downregulation also results in reduction of lipolysis [248]. Similarly, promotion of lipolysis and inhibition of lipid synthesis are dependent on SIRT2 and SIRT6, respectively [541,567]. On the other hand, lipogenesis is stimulated by SIRT4 [27]. Inflammatory processes in the adipose tissue are reduced by SIRT2 and SIRT3 that lower ROS levels, and by SIRT3 that stimulates cellular respiration and ameliorates crosstalk between macrophages and adipocytes [540,545,553,557]. Furthermore, mitochondrial functions are stimulated by SIRT1, SIRT3 and SIRT5 [373,519,520,553,563]. Thermogenesis is another local process that is affected by multiple Sirtuins in this tissue: SIRT1 stimulates it, while SIRT3 controls it in brown fat tissue, through deacetylation of thermoregulation proteins [519,520,547]. Finally, local insulin sensitivity is affected by expressions of SIRT1 and SIRT3, which reduce insulin resistance [508,555,556].

### 7.5. Skeletal Muscle

In the skeletal muscle, the AMPK-SIRT1 axis activates mitochondrial fatty acid oxidation, and it plays a role in the switch from glucose to fat metabolism [581]. AMPK senses changes in available nutrients and activates SIRT1, which subsequently activates PGC1α [582]; AMPK also directly phosphorylates PGC1α [582]. Exercise as well as diets can activate the AMPK/SIRT1/PGC1α axis [583,584,585,592,597,598,599]. Resveratrol and telmisartan activate the SIRT1/PGC1α and AMPK/SIRT1 pathways, to increase mitochondrial activity and aerobic capacity in muscle cells, respectively [589,590]. In addition, hyperinsulinemia increases SIRT1 expression in muscle tissue [610]. SIRT1 interacts with PI3K to enhance insulin sensitivity [600] and, via the PI3K/AKT pathway, promotes muscle growth while reducing atrophy in mice [605]. Furthermore, SIRT1 deacetylates and deactivates FOXO1 and FOXO3 [605]. Inhibition of FOXO1 prevents muscle atrophy and induces hypertrophy [603,604]. However, exercise induces muscle damages, in response to which AMPK increases, resulting in increased levels of FOXO1 and FOXO3a. These transcription factors interact with PGC1α to inhibit myoblast differentiation and muscle hypertrophy [606,607,608]. Activated AMPK also interacts with mTOR, to further inhibit proliferation [609]. The role of SIRT2 in metabolism, especially in the regulation of insulin sensitivity, remains a subject of debate. Indeed, potentially local and systemic effects of SIRT2 differ [471,602,612]. SIRT3 regulates oxidative stress and insulin signaling, although its effects differ at a local versus systemic level [563,564]. In addition, SIRT3 is involved in metabolism through mitochondria [617] and regulates the switch between fat and glucose metabolism [618]. SIRT3, together AMPK and PGC1α, form an axis that, upon activation, improves insulin sensitivity [626]. SIRT3 concentrations in the muscle tissue are modulated by diet and exercise [548,614]. SIRT4 regulates ATP levels [627] and, under specific circumstances, it switches the metabolism from fat to carbohydrates [628]. The relevant interaction partners of SIRT4 are AMPK and SIRT1 and, in fact, SIRT4 likely works upstream of AMPK in the AMPK/SIRT1 pathway [28]. SIRT5 does not seem to have a major role in metabolism in the muscle tissue [27]. SIRT6 seems instead to have a role in insulin sensitivity in muscle tissue, for which it interacts with AMPK and—mostly—AKT [388,629,630,632]. In addition, SIRT6 is vital to maintain muscle mass [633]. SIRT7 seems to regulate muscle protein degradation in older animals [635].

In the skeletal muscle, multiple Sirtuins influence metabolic pathways. SIRT1, SIRT2 and SIRT6 are involved in insulin sensitivity/resistance. However, conflicting findings are reported for SIRT2 and SIRT6 regarding their role in the stimulation or decrease of insulin sensitivity [389,473,614,633,634], and the cause of these inconsistencies has not been yet elucidated. A possible explanation may be related to differences between *in vitro* and *in vivo* studies, as well as to differences between systemic versus local SIRT6 deletion or overexpression [389]. SIRT1 decreases insulin resistance, likely through multiple routes since, alone, it is not sufficient to generate an effect [592,593,597,598,599,600,603]. Furthermore, SIRT3 maintains insulin action and can be upregulated through the compounds metformin and dihydromyricetin, to increase insulin sensitivity [621,627,628]. SIRT1 and SIRT4 play opposite roles in mitochondrial fatty acid oxidation, which is activated by SIRT1 and by SIRT4 knockdown, thus suggesting that SIRT4 suppresses this process in skeletal muscle tissue [28,583]. Finally, glucose metabolism—specifically glucose uptake—is influenced by SIRT2 and SIRT3, with SIRT2 downregulating glucose uptake in insulin-resistant cells and overexpression of SIRT3 increasing glucose uptake in muscle cells [614,621].

### 7.6. Heart

In the heart, SIRT1 is again the most studied among the Sirtuins. SIRT1 prevents cardiac hypertrophy, and protects against inflammation and metabolic dysregulation through interaction with PPARα and PGC1α [643,644]. However, the PPARα/SIRT1 pathway can also have detrimental effects upon inappropriate activation [647]. SIRT1 also regulates oxidative stress through regulation of p53, FOXO1, FOXO3 and FOXO4 [651,652,653,654]. The SIRT1 activator resveratrol leads to increased AMPK, PPARγ and PGC1α activity, while decreasing PPARα and p53 activity. In general, this outcome has positive effects on heart failure [656]. In addition, SIRT1 is involved in cardiac angiogenesis in response to cardiac hypertrophy by regulating AKT activity, which subsequently affects mTOR activity [660]. Finally, SIRT1 promotes autophagy and inhibits apoptosis after hypoxia in the heart via the AMPK/mTOR/AKT pathway [664], and protects against ER stress-induced apoptosis via the PERK/eIF2α-, ATF6/CHOP-, and IRE1α/JNK-mediated pathways [667]. The main function of SIRT2 is regulation microtubule stability by deacetylation, which protects the heart against hypertrophy [668]. To exert its function, SIRT2 interacts with and activates AMPK, while it represses NFAT transcription factors [673,674]. SIRT3 is involved in the regulation of mitochondrial deacetylation, thus preventing ventricular dysfunction and heart failure [675,676]. By activating AMPK and FOXO3a, SIRT3 exerts an anti-hypertrophic effect [678,689]. Furthermore, SIRT3, via FOXOs, reduces ROS levels, which subsequently block activity of AKT that may be either beneficial or detrimental [678]. SIRT4 is involved in regulating ROS levels by inhibiting MnSOD activity [691]. SIRT4 also prevents hypoxia-induced apoptosis during ischemic heart injury [692,694]. The main function of SIRT5 in the heart is desuccinylation of specific proteins that are mainly active in oxidative metabolism and oxidative phosphorylation [696]. Ultimately, this process influences cardiac hypertrophy, with SIRT5 knockout causing an increased hypertrophy and a systolic impairment [696]. SIRT6 is mainly involved in protection against ischemia, reperfusion injury, obesity and insulin resistance. In order to exert its effect, SIRT6 positively interacts with AMP/ATP, AMPK and FOXO3a [697,700]. In addition, SIRT6 and SIRT3 interact together to protect against diabetic cardiomyopathy [676]. Finally, SIRT6 prevents cardiac hypertrophy, by inhibiting the IGF-AKT pathway [698] and the NF-κB-dependent transcriptional activity [701]. SIRT7 is a stress-survival gene in cardiomyocytes [703,704,710], and it is involved in myocardial tissue repair through the growth factor B pathway [703]. Diet (CR) inhibits cardiovascular disease progression by increasing expression of SIRT4 and SIRT7.

In the heart, different Sirtuins share common local processes in which they function. ROS levels and oxidative stress response are affected by SIRT1, SIRT3, SIRT4, SIRT6 and SIRT7. Through activation of AKT, SIRT3 reduces ROS levels and thus oxidative stress, whereas SIRT4 increases oxidative stress through inhibition of the MnSOD activity [680,691,694]. SIRT1 regulates the oxidative stress response through a number of target genes, one of which—FOXO3—is also a target of SIRT6 to reduce oxidative stress [653,654,655,656,700]. Moreover, lack of SIRT7 results in a diminished resistance to oxidative stress [637,638]. Most Sirtuins in the heart prevent apoptosis of cardiomyocytes, during (SIRT7) or after (SIRT1, SIRT4) hypoxia [637,638,668,695]. SIRT1 can protect cardiomyocytes from apoptosis induced by ER-stress [669], and SIRT3, through melatonin, decreases apoptosis [692]. In addition, melatonin can modulate autophagy through SIRT3 signaling. Auto-phagy is generally regulated by SIRT3 and stimulated by SIRT1 through AMPK [668,685]. SIRT1 and SIRT3 also share additional local processes that they both affect. They prevent inflammation in the heart and affect mitochondrial functioning [645,683]: SIRT1 can induce mitochondrial dysfunction during a physiological stress response, whereas SIRT3 maintains mitochondrial function [649,682]. Moreover, through SIRT3 signaling, melatonin can alleviate mitochondrial dysfunction [692].

## 8. Conclusions

Generally, Sirtuins often interact with FOXOs, mTOR, AMPK, p53 and PPARs to exert their functions. However, these interactions are mostly described only for SIRT1 and SIRT3. Much has still to be discovered regarding the interaction partners of the other Sirtuins in a number of tissues.

Regarding the interventions to prevent metabolic dysfunctions, both CR and exercise are the most promising strategies that lead to a positive metabolic response from the Sirtuins in all tissues discussed here. Administration of the SIRT1 agonist resveratrol has a positive metabolic effect in all tissues, but the appropriate dosage differs per tissue. This observation results in the actual difficulty to develop an effective therapy at a systemic level. Therefore, the positive effects for the metabolism of an intervention in one tissue have to be weighed against any negative effects in other tissues.

This systematic review discusses, individually, a number of organs/metabolic tissues, whereas they are part of a whole interconnected system and function together as part of this system. Therefore, any intervention can have both local and systemic effects. Especially when a systemic administration of a drug induces adverse effects in another organ or at a systemic level, a local administration should be considered, when possible.

This systematic review summarizes the current knowledge about the roles of mammalian Sirtuins. However, it also shows that gaps exist about their functions, and that many links are missing regarding the Sirtuin-mediated modulation of their interaction partners in different metabolic tissues. Some of these directions, highlighted throughout this systematic review, may be prioritized in the future research, to understand better and subsequently modulate the human metabolic system to develop strategies helpful to fight against metabolic disorders.

## Figures and Tables

**Figure 1 biology-10-00194-f001:**
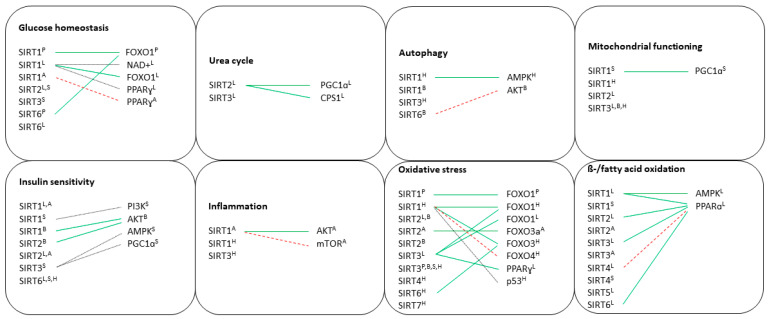
Metabolic processes where Sirtuins and their main targets are involved. GC1α, PPARα and AMPK are not indicated because activators, and not targets, of Sirtuins: GC1α and PPARα upregulate SIRT5 levels, whereas AMPK downregulates SIRT5 levels. Green lines, activatory interactions; red lines, inhibitory interactions; black lines, unspecified interactions. P, pancreas; L, liver; B, brain; A, adipose tissue; S, skeletal muscle; H, heart.

**Table 1 biology-10-00194-t001:** Overview of the role of Sirtuins in the pancreas.

Sirtuin	Expression and Localization	Effect of Sirtuin on Target Genes	Effect of Sirtuin on Local Processes	Involvement in Metabolic Diseases	Function in MetS (Effect on the Organism)	Effect of Dietary Influence on Sirtuin	Effect of Modulating Drugs on Sirtuin
Sirt1	Yes, β-cells	PGC1α (N/A) FOXO1 (↓) FOXA2 (↓) FOXO3a (N/A) PPARγ (N/A) AMPK (↑)	Glucose tolerance (↑) Insulin secretion (↑) β-cells proliferation (↑) Oxidative stress (↓)	T2DM	Positive	CR (↑) HFD (↓)	Resveratrol (↑) Artesunate (↑) GABA (↑) Fucoidan (↑)
Sirt2	Yes, RNA and cancer cells	N/A	N/A	N/A	N/A	N/A	N/A
Sirt3	Yes, β-cells	N/A	Inflammation (↓) Insulin secretion (↑) Oxidative stress (↓)	T2DM	Ambiguous	N/A	N/A
Sirt4	Yes, β-cells	N/A	Insulin secretion (↓)	N/A	Negative	CR (↓)	N/A
Sirt5	Yes, β-cells	N/A	Insulin signaling (↑↓) β-cell maintenance (↑↓)	T2DM	Ambiguous	N/A	N/A
Sirt6	Yes, β-cells	FOXO1 (↓)	Glucose tolerance (↑) Insulin signaling (↑)	N/A	Positive	N/A	N/A
Sirt7	Yes, exocrine glands and islets of Langerhans	N/A	N/A	N/A	N/A	N/A	N/A

**Table 2 biology-10-00194-t002:** Overview of the role of Sirtuins in the liver.

Sirtuin	Expression	Effect of Sirtuin on Target Genes	Effect of Sirtuin on Local Processes	Involvement in Metabolic Diseases	Function in MetS (Effect on the Organism)	Effect of Dietary Influence on Sirtuin	Effect of Modulating Drugs on Sirtuin
Sirt1	Yes	PGC1α (↑) PPARγ (N/A) FOXO1 (↑) AMPK (↑) PPARα (↑)	Gluconeogenesis (↑) Lipid metabolism (↑) Inflammation (↓) Insulin sensitivity (↑) β-oxidation (↑)	T2DM NAFLD	Positive	HFD (↑↓) CR (↓) Fasting (↑)	Metformin (↑) Resveratrol (↑) Maslinic acid (↑) Vitamin K (↑) Fisetin (↑) Fucoidan (↑) Mangiferin (↑) LB100 (↑) APE (↑) γ-mangostin (↑) Liraglutide (↑)
Sirt2	Yes	N/A	Insulin sensitivity (↑) Glucose uptake (↑) Glucose tolerance (↑)	N/A	Positive	N/A	N/A
Sirt3	Yes	PPARγ (N/A) FOXO1 (N/A) PPARα (N/A) PGC1α (N/A)	ROS scavenging (↑) Mitochondrial integrity (↑)	NAFLD	Positive	CR (↑) Ketogenic (↑) Fasting (↑)	Salvianolic acid B (↑) Berberine (↑)
Sirt4	Yes	SIRT1 (↓) PPARα (↓) mTOR (N/A)	Fatty acid oxidation (↓) Mitochondrial gene expression (↓)	N/A	Negative	N/A	3-Iodothyronamine (↓)
Sirt5	Yes	PGC1α (N/A) PPARα (N/A) AMPK (N/A)	Ketogenesis (↑) β-oxidation (↑)	NAFLD	Positive	Fasting (↑) High protein (↑)	N/A
Sirt6	Yes	SIRT1 (N/A) FOXO3a (N/A) PPARα (↑) ERRγ (↓)	Glycolysis (↓) Triglyceride synthesis (↓) Lipid metabolism (↓) Insulin sensitivity (↑) β-oxidation (↑)	NAFLD	Positive	N/A	Rolipram (↑) 3-Iodothyronamine (↑)
Sirt7	Yes	N/A	Triglyceride synthesis (↓)	NAFLD	Ambiguous	N/A	N/A

**Table 3 biology-10-00194-t003:** Overview of the role of Sirtuins in the brain.

Sirtuin	Expression and Localization	Effect of Sirtuin on Target Genes	Effect of Sirtuin on Local Processes	Involvement in Metabolic Diseases	Function in MetS (Effect on the Organism)	Effect of Dietary Influence on Sirtuin	Effect of Modulating Drugs on SIRTUIN
Sirt1	Yes, hypothalamus (POMS and AgRP neurons) and hippocampus	p53 (N/A) AMPK (N/A) PPARγ (↑) PGC1α (↑) mTOR (↓) FOXO3a (N/A)	Insulin sensitivity (↓) Systemic fat accumulation (↑) Inflammation (↓)	TD2M Ischemic injury	Negative	Fasting (↑) CR (↓) Ketosis (↑) Low protein (↑)	Resveratrol (↑) SRT1720 (↑)
Sirt2	Yes	N/A	Insulin sensitivity (↓) Ischemic injury (↓)	N/A	Negative	N/A	N/A
Sirt3	Yes, microglia, hippocampus and substantia nigra	FOXO3a (↑) PPARγ (↑) PGC1α (↑)	Mitochondrial biogenesis (↑) ROS reduction (↑)	Ischemic injury TD2M	Positive	Ketosis (↑)	Honokiol (↑)
Sirt4	Yes	Inhibited by mTOR	N/A	N/A	N/A	Glucose (↑)	N/A
Sirt5	Yes	N/A	Ischemic injury (↓)	N/A	Positive	CR (↑)	N/A
Sirt6	Yes, uniformly expressed	N/A	Apoptosis and cell survival (↑)	N/A	Ambiguous	CR (↑) Glucose (↑)	N/A
Sirt7	Yes	N/A	Neurogenesis (↑)	N/A	N/A	N/A	N/A

**Table 4 biology-10-00194-t004:** Overview of the role of Sirtuins in the adipose tissue.

Sirtuin	Expression and Localization	Effect of Sirtuin on Target Genes	Effect of Sirtuin on Local Processes	Involvement in Metabolic Diseases	Function in MetS (Effect on the Organism)	Effect of Dietary Influence on Sirtuin	Effect of Modulating Drugs on Sirtuin
Sirt1	Yes, brown and white adipose tissues	PPARδ (N/A) PPARγ (↓) mTOR (↓) FOXO1 (↑) Adiponectin (↑) AMPK (N/A)	Insulin resistance (↓) Adipogenesis (↓) Lipolysis (↑) Inflammation (↓) Mitochondrial function (↑) Glucose homeostasis (↑) Thermogenesis (↑)	Obesity	Positive	CR (↑) HFD (↓)	Resveratrol (↑) Spirulina-maxima (↑) Gallic acid (↑) Coumestrol (↑) *P. grandifloras* (↑)
Sirt2	Yes, brown and white adipose tissues	FOXO3a (↑) FOXO1 (↑) PGC1α (↑)	Adipogenesis (↓) ROS levels (↓) Lipolysis (↑)	Obesity	Positive	CR (↑) Fasting (↑)	N/A
Sirt3	Yes, brown and white adipose tissues	PGC1α (N/A) AMPK (N/A) PPARγ (N/A)	Insulin resistance (↓) Thermogenesis (↑) Cellular respiration (↑) ROS levels (↓) Lipogenesis (↓)	Obesity NALFD	Positive	CR (↑) Fasting (↑)	*P. grandifloras* (↑)
Sirt4	Yes, white adipose tissue	N/A	Lipogenesis (↑)	Obesity	Negative	HFD (↑)	N/A
Sirt5	Yes	N/A	N/A	Obesity	Ambiguous	N/A	N/A
Sirt6	Yes, brown and white adipose tissues	PPARγ (↓) FOXO1 (↑) AMPK (↑)	Glucose tolerance (↑) Insulin secretion (↑)	Obesity Lipidemia	N/A	N/A	Metformin (↑) Rolipram (↑) 3-Iodothyronamine (↑)
Sirt7	Yes	SIRT1 (↓)	Adipogenesis (↑)	Obesity	Ambiguous	N/A	N/A

**Table 5 biology-10-00194-t005:** Overview of the role of Sirtuins in the skeletal muscle.

Sirtuin	Expression	Effect of Sirtuin on Target Genes	Effect of Sirtuin on Local Processes	Involvement in Metabolic Diseases	Function in MetS (Effect on the Organism)	Effect of Dietary Influence on Sirtuin	Effect of Modulating Drugs on Sirtuin
Sirt1	Yes	PGC1α (↑) SIRT3 (↑) AMPK (N/A) FOXO1 (↓) FOXO3 (↓) PI3K (↑)	Mitochondrial activity (↑) Aerobic capacity (↑) Insulin resistance (↓) Fatty acid oxidation (↑)	T2DM	Positive	CR (↑) HFD (↓)	Resveratrol (↑) Telmisartan (↑)
Sirt2	Yes	N/A	Glucose uptake (↓)	N/A	Negative	N/A	N/A
Sirt3	Yes	AMPK (N/A) PGC1α (N/A) FOXO1 (↑)	Energy expenditure (↑) Respiratory rates (↓) Insulin resistance (↓) ROS levels (↓) Glucose uptake (↑)	T2DM	Positive	CR (↑) Fasting (↑) HFD (↓)	Metformin (↑)
Sirt4	Yes	AMPK (↓)	ATP levels (↑) Fatty acid oxidation (↓) Cellular respiration (↓)	N/A	Negative	N/A	N/A
Sirt5	Yes	N/A	N/A	N/A	N/A	N/A	N/A
Sirt6	Yes	AMPK (↑)	Insulin sensitivity (?)	N/A	Ambiguous	N/A	N/A
Sirt7	Yes	N/A	N/A	N/A	N/A	CR (↓)	N/A

**Table 6 biology-10-00194-t006:** Overview of the role of Sirtuins in the heart.

Sirtuin	Expression	Effect of Sirtuin on Target Genes	Effect of Sirtuin on Local Processes	Involvement in Metabolic Diseases	Function in MetS (Effect on the Organism)	Effect of Dietary Influence on Sirtuin	Effect of Modulating Drugs on Sirtuin
Sirt1	Yes	PGC1α (↑) FOXO1 (↑) FOXO3 (↑) FOXO4 (↑) p53 (↓) AMPK (↑) PGC1α (↑) PPARγ (↑) PPARα (↓)	Inflammation (↓) Oxidative stress (↓) Mitochondrial function (↓) Apoptosis (↓) Autophagy (↓)	Hypertrophy	Positive	CR (↑)	Resveratrol (↑)
Sirt2	Yes	AMPK (↑)	Microtubule activity (↓)	Hypertrophy T2DM	Positive	CR (↑)	N/A
Sirt3	Yes	AMPK (↑) SIRT6 (↑)	Oxidative stress (↓) Inflammation (↓) Apoptosis (↓) Mitochondrial function (↑) Autophagy	Hypertrophy	Positive	CR (↑) HFD (↓)	Melatonin (↑) Resveratrol (↑)
Sirt4	Yes	N/A	Oxidative stress (↑) Apoptosis (↓)	Hypertrophy	Ambiguous	CR (↑)	N/A
Sirt5	Yes	N/A	N/A	Hypertrophy	Positive	N/A	N/A
Sirt6	Yes	AMPK (↑) SIRT3 (↑)	Oxidative stress (↓) Insulin resistance (↓)	Ischemia T2DM Hypertrophy	Positive	HFD (↓)	N/A
Sirt7	Yes	p53 (↓)	Apoptosis (↓) Oxidative stress (↓)	Hypertrophy	Positive	CR (↑)	N/A

**Table 7 biology-10-00194-t007:** Overview of the role of Sirtuins in metabolic disorders.

Sirtuin	Effect in Organs and Disease Progression (P, Positive; N, Negative; N/A, Not Available; AM, Ambiguous)	Effect of Dietary Influence and Organs	Effect of Modulating Drugs	Role in Metabolic Diseases
Sirt1	Pancreas (P) Liver (P) Adipose tissue (P) Brain (N) Muscle (P) Heart (P)	CR (↑)—heart, muscle, adipose tissue, pancreas HFD (↓)—muscle, adipose tissue, pancreas HFD (↑↓)—liver CR (↑↓)—liver Fasting (↑)—brain CR (↓)—brain Ketosis (↑)—brain	Resveratrol (↑) Artesunate (↑) GABA (↑) Metformin (↑) Maslinic acid (↑) Vitamin K (↑) Fisetin (↑) Fucoidan (↑) Mangiferin (↑) Spirulina-maxima (↑) SRT1720 (↑) Telmisartan (↑)	Hypertrophy Obesity NAFLD T2DM Ischemic injury
Sirt2	Pancreas (N/A) Liver (P) Adipose tissue (P) Brain (N) Muscle (N) Heart (P)	CR (↑)—heart, adipose tissue Fasting (↑)—adipose tissue	N/A	Obesity T2DM Hypertrophy
Sirt3	Pancreas (AM) Liver (P) Adipose tissue (P) Brain (P) Muscle (P) Heart (P)	CR (↑)—heart, muscle, adipose tissue, liver, pancreas HFD (↓)—heart, muscle, pancreas Fasting (↑)—muscle, adipose tissue, liver Ketosis (↑)—brain, liver	Melatonin (↑) Resveratrol (↑) Metformin (↑) Honokiol (↑) Salvianolic acid B (↑) Berberine (↑)	Hypertrophy Obesity NAFLD T2DM Ischemic injury
Sirt4	Pancreas (N) Liver (N) Adipose tissue (N) Brain (N/A) Muscle (N) Heart (AM)	HFD (↑)—adipose tissue Glucose (↑)—brain CR (↓)—pancreas	3-Iodothyronamine (↓)	Hypertrophy Obesity
Sirt5	Pancreas (AM) Liver (P) Adipose tissue (AM) Brain (P) Muscle (N/A) Hear (P)	Fasting (↑)—liver High protein (↑)—liver CR (↑)—brain	N/A	Hypertrophy Obesity NAFLD T2DM
Sirt6	Pancreas (P) Liver (N/A) Adipose tissue (N/A) Brain (AM) Muscle (AM) Heart (P)	HFD (↓)—heart CR (↑)—brain Glucose (↑)—brain	Metformin (↑) Rolipram (↑) 3-Iodothyronamine (↑)	Ischemia T2DM Hypertrophy Obesity Lipidemia NAFLD
Sirt7	Pancreas (N/A) Liver (AM) Adipose (N/A) Brain (N/A) Muscle (N/A) Heart (P)	CR (↓)—muscle CR (↑)—heart	N/A	Hypertrophy Obesity NAFLD

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
