# Peer review of "Sirtuins-Mediated System-Level Regulation of Mammalian Tissues at the Interface between Metabolism and Cell Cycle: A Systematic Review"

_biology, 2021, doi:10.3390/biology10030194_

Round 1

Reviewer 1 Report

In this review entitled “Sirtuins-Mediated System-Level Regulation of Mammalian Tissues at the Interface between Metabolism and Cell Cycle: A Systematic Review”, the authors have done a substantial data collection and organized information regarding the role/involvement of sirtuins in many aspects of mammalian metabolism in specific tissues. Using a PubMed search, the authors’ query yielded a conspicuous number of results that have been reported and correlated where possible.

Overall the work is interesting, however I have a number of conceptual criticisms that in my opinion need to be resolved and some minor points to be addressed.

Major points

1.The research profile authors employed "(SIRT1 OR SIRT2 OR SIRT3 OR SIRT4 OR SIRT5 OR SIRT6 OR SIRT7) AND metabolism AND (pancreas OR liver OR heart OR muscle OR adipose tissue OR brain)" allowed to obtain a very detailed picture of the field. However, it overlooked the role of sirtuins as histone deacetylases, the enzymatic activity for which they were initially discovered. Not exploring this aspect represents a weak point of the work, especially in the light of the metabolic involvement of sirtuins. In fact, the use of NAD+, essential for their activity, implies metabolic conditions that usually are antithetical to the availability of AcetylCoA (precursor of histone acetylation). Therefore, gene expression is deeply influenced by the enzymatic functionality of sirtuins, as well. In my opinion this point must be implemented at least in the initial part of the work.

2.Given the large amount of data presented, and although summarized with 7 tables, the authors should further facilitate the reader’s understanding by the use of figures or cartoons, possibly by overlapping the main metabolic pathways reported in the review (pathway of glycolysis, β-oxidation, via insulin, etc. etc.) with the specific sirtuin’s function involved. This would make the authors’ effort more functional.

3.The discussion needs revision because, as such, it is just a summary of all the data rather than a real discussion of the relevant issues related to the mechanisms, pathways, pathologies and tissues. Possibly, dividing it in paragraphs of different sirtuins or metabolisms or tissues would help.

Minor points

line 52 - Rephrase: Sirtuins influence and are affected by metabolism

line 76 - there is a nomenclature mistake: sirtuins are not “silent information regulators”, this category of proteins has been found in yeast (SIR1, 2 3, 4) and only Sir2 is a histone deacetylase i.e. a sirtuin. The acronym SIRT stands for SIR-TWO (family) 1, 2….7!!!

line 96 - histone transferase must be histone acetyl transferase

Line 103 - changes to: 2-O acetyl-ribose

Line 128 - might be relevant to talk about tubulin deacetylation activity during mitosis; this is not properly mentioned, please include a sentence about it.

Lines 183 to 207: the authors describe in detail the metabolic pathway compared to others. This should be shortened

Line 208 - “Pathways available in an organization ……” this sentence is not clear, it should be rephrased

line 296 - Metabolic regulation also influences gene expression and should be discussed

Section 3.2 - Epigenetic modification precursors are not limited to AcCoA but are also ATP, SAM, NAD+, α-ketoglutarate etc. Principal precursors should be mentioned.

Line 623 - ref 170 is indicated, however a few words must be added about the mechanism

Line 688 – also for these refs some details must be added

Line 793 - here a diagram indicating the relationships would be helpful

Line 822 - Sir2 is the SIRT1 ortholog

Reviewer 2 Report

This is a comprehensive review on metabolic pathways and Sirtuin activity. The review is very broad covering many aspects of metabolism and cell cycle before delving into Sirtuin biology. If this was the aim of the authors, it is very detailed and well written. If the authors aim to have a more concise review on Sirtuins, some of the aforementioned sections could be reduced significantly. The review lists the results of many studies, but really fails to discuss or link these together in a meaningful way, and as such is just a list of results from various studies. There is also conflicting reports, as highlighted in the adipose SIRT1 section and muscle SIRT6 section. The authors need to discuss these conflicting studies and interpret them and present their opinion. More discussion and interpretation of the studies overall would benefit the review, where presently it is limited to a paragraph at the end of each section. • Line 99: HAT = histone acetyltransferases. • Line 732 & 744-745: conflicting PPARy tissue expression. • Line 1446: activating SIRT1 = healthy adipose expansion, ie hyperplasia, yet at Line 1441 & 1450 SIRT1 = inhibition of adipogenesis. This is conflicting. • Similarly line 1456 = reduced SIRT1 activity = increase in MetS, yet SIRT1 KO = improves metabolic parameters, but at Line 1507: upregulation of SIRT1 = improvements in glucose handling etc. In addition Line 1467: SIRT1 Knockdown = inflammation which is not beneficial?? • These findings are not accurately reflected in the corresponding table. Eg. Sirt1 = reduced PPARy (which is generally associated with poor metabolic outcomes such as insulin resistance) yet function in MetS is positive?? • There are conflicting results in regards to SIRT6 in Skeletal muscle: line 1807 = increase in SIRT6 improves insulin sensitivity and is up during GTT for increased glucose uptake. However, line 1810, Sirt6 deficiency = increased insulin signalling and glucose uptake. So is SIRT6 good or bad? Overexpression of Sirt6 (line 1822) = diabetes risk and hyperglacamia, yet SIRT6 KO = reduced muscle mass etc (Line 1816) • Minor spelling and grammatical errors throughout manuscript
